# Efficiently Access Diffusion Fisher: Within the Outer Product Span Space

**Fangyikang Wang** [1] [*] [‡]   **Hubery Yin** [2] [*]   **Shaobin Zhuang** [3] [‡]   **Huminhao Zhu** [1]   **Yinan Li** [1]   **Lei Qian** [1]
**Chao Zhang** [1]   **Hanbin Zhao** [1]   **Hui Qian** [1]   **Chen Li** [2]

## Abstract

Recent Diffusion models (DMs) advancements have explored incorporating the second-order diffusion Fisher information (DF), defined as the negative Hessian of log density, into various downstream tasks and theoretical analysis. However, current practices typically approximate the diffusion Fisher by applying auto-differentiation to the learned score network. This black-box method, though straightforward, lacks any accuracy guarantee and is time-consuming. In this paper, we show that the diffusion Fisher actually resides within a space spanned by the outer products of score and initial data. Based on the outer-product structure, we develop two efficient approximation algorithms to access the trace and matrix-vector multiplication of DF, respectively. These algorithms bypass the auto-differentiation operations with time-efficient vector-product calculations. Furthermore, we establish the approximation error bounds for the proposed algorithms. Experiments in likelihood evaluation and adjoint optimization demonstrate the superior accuracy and reduced computational cost of our proposed algorithms. Additionally, based on the novel outer-product formulation of DF, we design the first numerical verification experiment for the optimal transport property of the general PF-ODE deduced map.

## 1. Introduction

The emerging diffusion models (DMs) (Sohl-Dickstein et al., 2015; Ho et al., 2020; Song & Ermon, 2019; Song et al., 2020), generating samples of data distribution from initial noise by learning a reverse diffusion process, have been proven to be an effective technique for modeling data distri-

bution, especially in generating high-quality images (Nichol et al., 2022; Dhariwal & Nichol, 2021a; Saharia et al., 2022; Ramesh et al., 2022; Rombach et al., 2022; Ho et al., 2022). The training process of DMs can be seen as employing a neural network to match the first-order diffusion score $\nabla_{\boldsymbol{x}} \log q_t(\boldsymbol{x})$ at varying noise levels.

There has been a growing trend to recognize the importance of the diffusion Fisher (DF) in DMs, defined as the negative Hessian of the diffused distributions' log density, $-\nabla_{\boldsymbol{x}}^2 \log q_t(\boldsymbol{x})$. The DF provides valuable second-order information on DMs and plays a crucial role in downstream tasks, e.g., likelihood evaluation (Lu et al., 2022a; Zheng et al., 2023), adjoint optimization (Pan et al., 2023a;b; Blasingame & Liu, 2024), and the optimal transport analysis of DMs (Zhang et al., 2024a). It is straightforward to access DFs by applying auto-differentiation to the learned first-order diffusion score. Nevertheless, the repeated auto-differentiation operations would be time-consuming for large-scale tasks, even when incorporating the Vector-Jacobian-product (VJP) and Hutchinson's estimation technique (Song & Lai, 2024). Recently, certain research efforts investigated the time-evolving property of DM and demonstrated that the DF can be explicitly expressed in terms of the score and its covariance matrix (Benton et al., 2024; Lu et al., 2022a). However, these results predominantly concentrate on the theoretical aspects and fail to enable efficient access to the DF.

In this paper, we derive a novel form for the diffusion Fisher, which is composed of weighted outer-product sums. This innovative formulation uncovers that the diffusion Fisher invariably lies within the span of a set of outer-product bases. Notably, these bases solely depend on the initial data and the noise schedule. Leveraging this outer-product structure, we devise two efficient approximation algorithms to access the trace and matrix-vector multiplication of DF, respectively. Moreover, we design the first numerical verification experiment to investigate the optimal transport property of the diffusion-ODE deduced map under diverse types of initial data and noise schedules. The main contributions of our paper are listed as follows:

- We derive an analytical formulation of the diffusion Fisher, which is the weighted summation of the outer

[1]Zhejiang University. [2]WeChat Vision, Tencent Inc. [3]Shanghai Jiao Tong University. [*]Equal contribution. [‡]Work done as interns at WeChat Vision, Tencent Inc. Correspondence to: Chao Zhang <zczju@zju.edu.cn>.

*Proceedings of the 42nd International Conference on Machine Learning*, Vancouver, Canada. PMLR 267, 2025. Copyright 2025 by the author(s).

product, under the setting when the initial distribution is a sum of Dirac. The derivation process involves applying the consecutive partial differential chain rule to the data-dependent form of marginal distributions. Subsequently, we extend this formulation to a more general setting. In this general case, we only assume that the initial distribution has a finite second moment. These novel formulations reveal that the diffusion Fisher invariably lies within the span of a set of outer-product bases.

- Based on the outer-product structure of DF, we propose two efficient approximation algorithms, each tailored to access the trace and matrix-vector multiplication of the DF, respectively. When it comes to evaluating the trace of the diffusion Fisher, we introduce a parameterized network to learn the trace, significantly reducing the time complexity of trace evaluation from quadratic to linear w.r.t. the dimension. In situations where we need to access the matrix-vector multiplication of the diffusion Fisher, we present a training-free method that simplifies the complex linear transformation into several simple vector-product calculations. Furthermore, we rigorously establish the approximation error bounds for these two algorithms.

- Utilizing the novel outer-product formulation of DF, we derive a corollary for the optimal transport (OT) property of the diffusion-ODE deduced map. Subsequently, we design the first numerical experiment to examine the OT property of any general diffusion ODE deduced map based on the corollary. Our numerical experiments indicate that for all commonly employed noise schedules (VE (Ho et al., 2020), VP (Song & Ermon, 2019), sub-VP (Song et al., 2020), and EDM (Karras et al., 2022)), the diffusion ODE map exhibits the OT property when the initial data follows a single Gaussian distribution or has an affine form. Conversely, it does not exhibit this property when the initial data is non-affine. These numerical results not only validate the conclusions presented in (Khrulkov et al., 2023; Zhang et al., 2024a; Lavenant & Santambrogio, 2022), but also open up new avenues for the exploration of more general cases.

We evaluate our DF access algorithms on likelihood evaluation and adjoint optimization tasks. The empirical results demonstrate our DF methods' enhanced accuracy and reduced time cost.

## 2. Preliminaries

**Notation**. The Euclidean norm over $\mathbb{R}^d$ is denoted by $\|\cdot\|$, and the Euclidean inner product is denoted by $\langle\cdot|\cdot\rangle$. Throughout, we simply write $\int g$ to denote the integral with respect

to the Lebesgue measure: $\int g(x)\mathrm{d}x$. When the integral is with respect to a different measure $\mu$, we explicitly write $\int g\mathrm{d}\mu$. When clear from context, we sometimes abuse notation by identifying a measure $\mu$ with its Lebesgue density. We also use $\delta(\cdot)$ to denote the Dirac Delta function. For a vector $\boldsymbol{v} \in \mathbb{R}^d$, we denote the $d \times d$ matrix $\boldsymbol{v}\boldsymbol{v}^\top$ as the outer-product of $\boldsymbol{v}$.

### 2.1. Diffusion Models and Diffusion SDEs

Suppose that we have a d-dimensional random variable $\boldsymbol{x}_0 \in \mathbb{R}^d$ following an unknown target distribution $q_0(\boldsymbol{x}_0)$. Diffusion Models (DMs) define a forward process $\{\boldsymbol{x}_t\}_{t\in[0,T]}$ with $T > 0$ starting with $\boldsymbol{x}_0$, such that the distribution of $\boldsymbol{x}_t$ conditioned on $\boldsymbol{x}_0$ satisfies

$$q_{t|0}(\boldsymbol{x}_t|\boldsymbol{x}_0) = \mathcal{N}(\boldsymbol{x}_t; \alpha(t)\boldsymbol{x}_0, \sigma^2(t)\mathbf{I}), \quad (1)$$

where $\alpha(\cdot), \sigma(\cdot) \in \mathcal{C}([0,T], \mathbb{R}^+)$ have bounded derivatives, and we denote them as $\alpha_t$ and $\sigma_t$ for simplicity. The choice for $\alpha_t$ and $\sigma_t$ is referred to as the noise schedule of a DM. According to (Kingma et al., 2021; Karras et al., 2022), with some assumption on $\alpha(\cdot)$ and $\sigma(\cdot)$, the forward process can be modeled as a linear SDE which is also called the Ornstein–Uhlenbeck process:

$$\mathrm{d}\boldsymbol{x}_t = f(t)\boldsymbol{x}_t\mathrm{d}t + g(t)\mathrm{d}B_t, \quad (2)$$

where $B_t$ is the standard d-dimensional Brownian Motion (BM), $f(t) = \frac{\mathrm{d}\log\alpha_t}{\mathrm{d}t}$ and $g^2(t) = \frac{\mathrm{d}\sigma_t^2}{\mathrm{d}t} - 2\frac{\mathrm{d}\log\alpha_t}{\mathrm{d}t}\sigma_t^2$. Under some regularity conditions, the above forward SDE equation 2 have a reverse SDE from time $T$ to 0, which starts from $\boldsymbol{x}_t$ (Anderson, 1982):

$$\mathrm{d}\boldsymbol{x}_t = \left[f(t)\boldsymbol{x}_t - g^2(t)\nabla_{\boldsymbol{x}_t}\log q(\boldsymbol{x}_t, t)\right]\mathrm{d}t + g(t)\mathrm{d}\tilde{B}_t, \quad (3)$$

where $\tilde{B}_t$ is the reverse-time Brownian motion and $q(\boldsymbol{x}_t, t)$ is the single-time marginal distribution of the forward process. In practice, DMs (Ho et al., 2020; Song et al., 2020) use $\boldsymbol{\varepsilon}_\theta(\boldsymbol{x}_t, t)$ to estimate $-\sigma(t)\nabla_{\boldsymbol{x}_t}\log q(\boldsymbol{x}_t, t)$ and the parameter $\theta$ is optimized by the following loss:

$$\mathcal{L} = \mathbb{E}_t\left\{\lambda_t\mathbb{E}_{x_0,x_t}\left[\|s_\theta(x_t, t) - \nabla_{x_t}\log p(x_t, t|x_0, 0)\|^2\right]\right\}, \quad (4)$$

where $s_\theta$ represents the parameterized score, i.e., $s_\theta(\boldsymbol{x}_t, t) = -\frac{\boldsymbol{\varepsilon}_\theta(\boldsymbol{x}_t, t)}{\sigma_t}$. This familiar parameterization is called $\epsilon$-prediction. There are also $\boldsymbol{y}$-prediction and $\boldsymbol{v}$-prediction (Salimans & Ho, 2022). The corresponding loss is equal to replace the term $|\epsilon - \boldsymbol{\varepsilon}_\theta(\boldsymbol{x}_t, t)|$ with $\frac{\alpha_t}{\sigma_t}|\boldsymbol{x}_0 - \bar{\boldsymbol{y}}_\theta(\boldsymbol{x}_t, t)|$ and $|\alpha_t\epsilon - \sigma_t\boldsymbol{x}_0 - \boldsymbol{v}_\theta(\boldsymbol{x}_t, t)|$. The learned $\boldsymbol{\varepsilon}_\theta(\boldsymbol{x}_t, t)$ can be also transformed to a $\boldsymbol{y}$-prediction form by $\bar{\boldsymbol{y}}_\theta(\boldsymbol{x}_t, t) = \frac{\boldsymbol{x}_t - \sigma_t\boldsymbol{\varepsilon}_\theta(\boldsymbol{x}_t, t)}{\alpha_t}$.

### 2.2. Diffusion Models Inference as Neural ODE

It is noted that the reverse diffusion SDE in equation 3 has an associated probability flow ODE (PF-ODE, also called diffu-

sion ODE), which is a deterministic process that shares the same single-time marginal distribution (Song et al., 2020):

$$d\boldsymbol{x}_t = \left[ f(t)\boldsymbol{x}_t - \frac{1}{2} g^2(t) \nabla_{\boldsymbol{x}_t} \log q_t(\boldsymbol{x}_t, t) \right] dt. \quad (5)$$

By replacing the score function in equation 5 with the noise predictor $\boldsymbol{\varepsilon}_\theta$, the inference process of DMs can be constructed by the following neural ODE:

$$\frac{d\boldsymbol{x}_t}{dt} = \boldsymbol{h}_\theta(\boldsymbol{x}_t, t) := f(t)\boldsymbol{x}_t + \frac{g^2(t)}{2\sigma_t} \boldsymbol{\varepsilon}_\theta(\boldsymbol{x}_t, t), \quad (6)$$

where initial $\boldsymbol{x}_T \sim \mathcal{N}\left(\boldsymbol{0}, \sigma_T^2 \boldsymbol{I}\right)$.

### 2.3. Fisher Information in Diffusion Models

The diffusion Fisher information matrix in DMs is defined as the negative Hessian of the marginal diffused log-density function, which takes the following matrix-valued form (Song et al., 2021; Song & Lai, 2024):

$$\boldsymbol{F}_t(\boldsymbol{x}_t, t) := -\frac{\partial^2}{\partial \boldsymbol{x}_t^2} \log q_t(\boldsymbol{x}_t, t) \quad (7)$$

The current technique typically approximately accesses the diffusion Fisher by accessing the scaled Jacobian matrix of the learned score estimator network $\boldsymbol{\varepsilon}_\theta$:

$$
\begin{aligned}
\boldsymbol{F}_t(\boldsymbol{x}_t, t) &= -\frac{\partial}{\partial \boldsymbol{x}_t} \left( \frac{\partial}{\partial \boldsymbol{x}_t} \log p(x_t, t) \right) \\
&\approx -\frac{\partial}{\partial \boldsymbol{x}_t} \left( -\frac{\boldsymbol{\varepsilon}_\theta(\boldsymbol{x}_t, t)}{\sigma_t} \right) = \frac{1}{\sigma_t} \frac{\partial \boldsymbol{\varepsilon}_\theta(\boldsymbol{x}_t, t)}{\partial \boldsymbol{x}_t}
\end{aligned} \quad (8)
$$

The full diffusion Fisher matrix within DMs cannot be obtained due to dimensional constraints. For instance, the Stable Diffusion-1.5 model (Rombach et al., 2022) features a latent dimension of $d = 4 \times 64 \times 64 = 16384$, resulting in a diffusion Fisher matrix of $16384 \times 16384$. Fortunately, for applications that only need to access the trace or multiplication of diffusion Fisher, it is feasible to use the Vector-Jacobian-product (VJP) to access diffusion Fisher. For any $d$-dimensional vector $\boldsymbol{v}$, the approximation of $\boldsymbol{v}$ left multiplied by $\boldsymbol{F}_t(\boldsymbol{x}_t, t)$ using VJP is as follows:

$$
\begin{aligned}
\text{(VJP)} \qquad \boldsymbol{v}^\top \boldsymbol{F}_t(\boldsymbol{x}_t, t) &\approx \frac{1}{\sigma_t} \boldsymbol{v}^\top \frac{\partial \boldsymbol{\varepsilon}_\theta(\boldsymbol{x}_t, t)}{\partial \boldsymbol{x}_t} \\
&= \frac{1}{\sigma_t} \frac{\partial \left[ \langle \boldsymbol{\varepsilon}_\theta(\boldsymbol{x}_t, t) | \boldsymbol{v} \rangle \right]}{\partial \boldsymbol{x}_t}
\end{aligned} \quad (9)
$$

The VJP is a time-consuming process due to its requirement for gradient calculations within the neural network. In addition, empirical evidence from synthetic distributions, as demonstrated in Lu et al. (2022a), shows that the approximation results from the VJP significantly deviate from the true underlying diffusion Fisher. To our knowledge, there

| Access | Methods | Theoretical Time-cost | Practical Time-cost (s) | Theoretical Error Bound |
|---|---|---|---|---|
| Trace | VJP (eq. 15) | $c_1 d + c_2 d^2$ | 2195.48 | ✗ |
| | DF-TM (**Ours**) | $2c_1 d$ | **0.072**$_{99\%\downarrow}$ | ✓(Prop. 7) |
| Operator | VJP (eq. 19) | $c_1 d + c_2 d$ | 0.155 | ✗ |
| | DF-EA (**Ours**) | $2c_1 d$ | **0.063**$_{59\%\downarrow}$ | ✓(Prop. 8) |

Table 1: Comparison of Vector-Jacobian-product (VJP) and our proposed efficient access methods in terms of per-iteration theoretical time-cost, practical time-cost, and approximation error bound. The theoretical time cost is based on assumptions of a network access cost of $c_1 d$ time and backpropagation on network cost of $c_2 d$ time ($c_2 \approx 4c_1$). The practical time-cost is tested using the SD-v1.5 model over 10k COCO prompts. Here, our VJP baseline calculates every element of the trace, and discussion on its approximation is deferred to Appendix C.5.

is no theoretical guarantee that the diffusion Fisher can be accurately accessed through the VJP. Moreover, the VJP fails to provide any theoretical insight into the diffusion Fisher.

## 3. Diffusion Fisher in Outer-Products Span

Accessing diffusion Fisher via the VJP as shown in equation 9 is straightforward, but it does not take advantage of any inherent structure of the diffusion Fisher. Though certain recent research efforts (Lu et al., 2022a; Benton et al., 2024) demonstrated that the DF can be explicitly expressed in terms of the score and its covariance matrix. Nevertheless, their formulations cannot directly facilitate efficient access to the diffusion Fisher. In this section, we derive a novel form for the diffusion Fisher, which is composed of weighted outer-product sums. We first conduct this under a simplified scenario, assuming that the initial distribution $q_0$ is a sum of Dirac distributions. Subsequently, we generalize this outer-product span form to a more general setting. Significantly, the outer-product span form of the diffusion Fisher obtained in both settings does not require any gradient calculations and depends solely on the initial data and the noise schedule. Notably, there exist fast methods for accessing the trace and matrix-vector multiplication of an outer-product matrix. This advantageous characteristic allows for the development of novel algorithms for efficiently accessing the diffusion Fisher. We start with a simple setting where we assume that the initial distribution is characterized as a sum of Dirac distributions composed of the set of samples in the dataset. If we suppose the dataset is denoted as $\{\boldsymbol{y}_i\}_{i=0}^N$, then the initial distribution follows

$$\text{(Dirac Setting)} \qquad q(\boldsymbol{x}, t)|_{t=0} = \frac{1}{N} \sum_{i=0}^N \delta(\boldsymbol{x} - \boldsymbol{y}_i), \quad (10)$$

where exists a $0 < \mathcal{D}_y < \infty$ such that $\|\boldsymbol{y}_i\| \leq \mathcal{D}_y$ holds true for every $i$. In this Dirac setting, we derive the following formulation of diffusion Fisher, which is a weighted outer-product sum devoid of gradients and composed solely of the

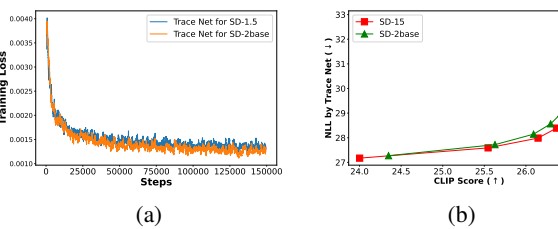

(a)    (b)

Figure 1: (a) The training loss of DF-TM for SD-1.5 and SD-2base. It demonstrates commendable convergence behavior. (b) The trade-off curve of NLL and Clip score of SD-1.5 and SD-2base across various guidance scales in [1.5, 2.5, ..., 12.5, 13.5]

initial distribution and the noise schedule.

*Proposition 1.* Defines $v_i(\boldsymbol{x}_t, t)$ as $\exp\left(-\frac{|\boldsymbol{x}_t - \alpha_t \boldsymbol{y}_i|^2}{2\sigma_t^2}\right) \in \mathbb{R}$ and $w_i(\boldsymbol{x}_t, t)$ as $\frac{v_i(\boldsymbol{x}_t, t)}{\sum_j v_j(\boldsymbol{x}_t, t)} \in \mathbb{R}$. If $q_0$ takes the form as in equation equation 10, the diffusion Fisher matrix of the diffused distribution $q_t$ for $t \in (0, 1]$ can be analytically formulated as follows:

$$\boldsymbol{F}_t(\boldsymbol{x}_t, t) = \frac{1}{\sigma_t^2}\boldsymbol{I} - \frac{\alpha_t^2}{\sigma_t^4}\left[\sum_i w_i \boldsymbol{y}_i \boldsymbol{y}_i^\top - \left(\sum_i w_i \boldsymbol{y}_i\right)\left(\sum_i w_i \boldsymbol{y}_i\right)^\top\right] \quad (11)$$

where we have simplified $w_i(\boldsymbol{x}_t, t)$ to $w_i$, as it does not lead to any confusion.

We also find that the $\sum_i w_i \boldsymbol{y}_i$ component in equation 11 can be effectively approximated by the trained score network in the form of $y$-prediction, as demonstrated in the following proposition.

*Proposition 2.* Given the diffusion training loss in equation 4, and if $q_0$ conforms to the form presented in equation 10, then the optimal $\bar{\boldsymbol{y}}_\theta(\boldsymbol{x}_t, t)$ can accurately estimate $\sum_i w_i \boldsymbol{y}_i$.

**The General Setting** We further extend the diffusion Fisher formulation in equation 11 to a more general setting, where we only assume that the initial distribution $q_0$ is a probability measure on $\mathbb{R}^d$ with finite second moments.

$$\text{(General Setting)} \qquad q_0 \in \mathcal{P}_2(\mathbb{R}^d), \qquad (12)$$

which means that $q_0 \in \mathcal{P}(\mathbb{R}^d)$, $\int_{\mathbb{R}^d} ||x||^2 q_0(x)\mathrm{d}x < \infty$. In this general setting, we derive the following form of diffusion Fisher, which is a weighted outer-product integral, which resides in a space spanned by an infinite set of outer products.

*Proposition 3.* Let us define $v(\boldsymbol{x}_t, t, \boldsymbol{y})$ as $\exp\left(-\frac{|\boldsymbol{x}_t - \alpha_t \boldsymbol{y}|^2}{2\sigma_t^2}\right) \in \mathbb{R}$ and $w(\boldsymbol{x}_t, t, \boldsymbol{y})$ as $\frac{v(\boldsymbol{x}_t, t, \boldsymbol{y})}{\int_{\mathbb{R}^d} v(\boldsymbol{x}_t, t, \boldsymbol{y})\mathrm{d}q_0(\boldsymbol{y})} \in \mathbb{R}$. If $q_0$ takes the form as in equation 12, the diffusion Fisher matrix of the diffused distribution $q_t$ for $t \in (0, 1]$ can be analytically formulated as follows:

$$\boldsymbol{F}_t(\boldsymbol{x}_t, t) = \frac{1}{\sigma_t^2}\boldsymbol{I} - \frac{\alpha_t^2}{\sigma_t^4}\left[\int w(\boldsymbol{y})\boldsymbol{y}\boldsymbol{y}^\top \mathrm{d}q_0 - \left(\int w(\boldsymbol{y})\boldsymbol{y}\mathrm{d}q_0\right)\left(\int w(\boldsymbol{y})\boldsymbol{y}\mathrm{d}q_0\right)^\top\right] \quad (13)$$

where we simply write $w(\boldsymbol{x}_t, t, \boldsymbol{y})$ as $w(\boldsymbol{y})$, as long as it does not lead to any confusion.

We further ascertain that the $\int w(\boldsymbol{y})\boldsymbol{y}\mathrm{d}q_0(\boldsymbol{y})$ component in equation 13 can be effectively approximated by the score network in the form of $y$-prediction, as demonstrated in the following proposition.

*Proposition 4.* Given the diffusion loss in equation 4, and if $q_0$ conforms to the form in equation 12, then the optimal $\bar{\boldsymbol{y}}_\theta(\boldsymbol{x}_t, t)$ can accurately estimate $\int w(\boldsymbol{y})\boldsymbol{y}\mathrm{d}q_0(\boldsymbol{y})$.

The derivation of the outer-product form of diffusion Fisher under the general setting is akin to the Dirac case, but in an integral form. For the remainder of the paper, we will focus on developing our method based on the Dirac setting diffusion Fisher. However, the same results can be naturally extended to the general setting.

## 4. Diffusion Fisher Trace Matching

The likelihood evaluation of DMs would require access to diffusion Fisher's trace. In this section, we introduce a network to learn the trace, thus facilitating effective likelihood evaluation in DMs.

**Log-Likelihood in DMs** Log-likelihood is a classic and significant metric for probabilistic generative models, extensively utilized for comparison between samples or models (Bengio et al., 2013; Theis et al., 2015). According to Chen et al. (2018); Song et al. (2021), the log-likelihood of samples generated by PF-ODE in equation 6 from DMs can be computed through a connection to continuous normalizing flows as follows:

$$\frac{\partial \log q_t(\boldsymbol{x}_t, t)}{\partial t} = -\mathrm{tr}\left(\frac{\partial}{\partial \boldsymbol{x}_t}\left(f(t)\boldsymbol{x}_t - \frac{1}{2}g^2(t)\partial_{\boldsymbol{x}_t}\log q_t(\boldsymbol{x}_t, t)\right)\right)$$
$$= -\mathrm{tr}\left(\left(f(t)\boldsymbol{I} - \frac{1}{2}g^2(t)\frac{\partial^2}{\partial \boldsymbol{x}_t^2}\log q_t(\boldsymbol{x}_t, t)\right)\right)$$
$$= -f(t)d - \frac{g^2(t)}{2}\mathrm{tr}\left(\boldsymbol{F}_t(\boldsymbol{x}_t, t)\right) \quad (14)$$

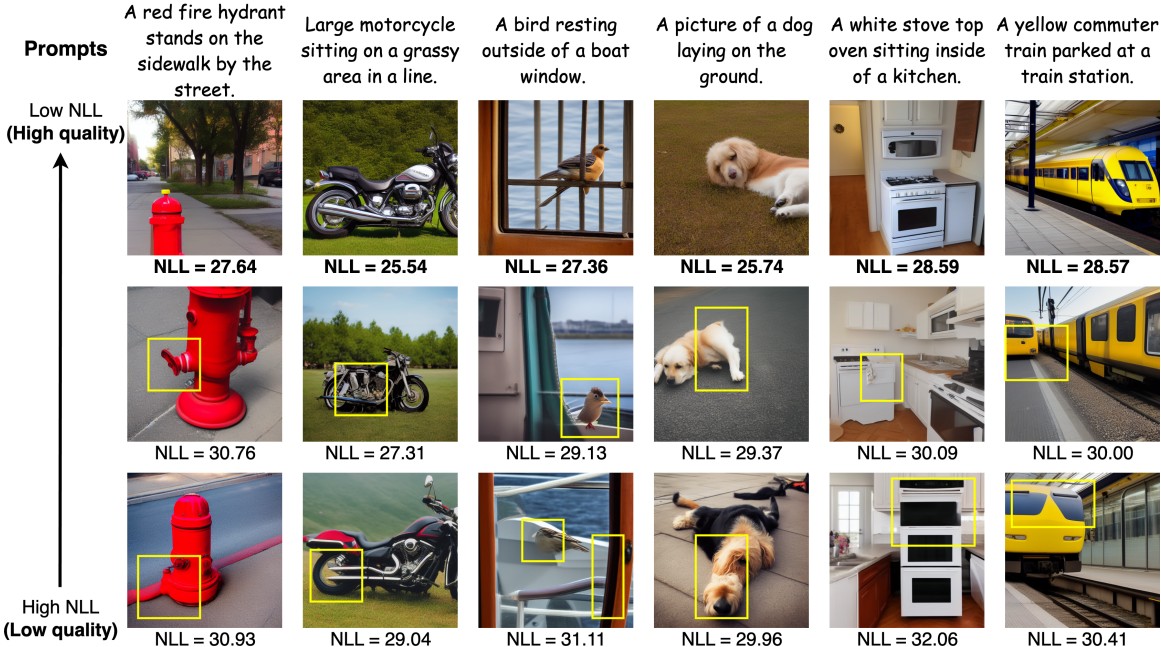

Figure 2: Our DF-TM method facilitates the effective evaluation of the NLL of generated samples with varying seeds. It can be demonstrated that a lower NLL signifies a region of higher possibility, thereby consistently indicating superior image quality.

where $\mathrm{tr}(\cdot)$ denotes the trace of a matrix, which is defined to be the sum of elements on the diagonal.

**Log-Likelihood Evaluation via VJP**  The current technique is only capable of conducting backpropagation of scalar value to the neural network. Therefore, the VJP in equation 9 cannot directly calculate the trace of the diffusion Fisher. The VJP must iterate through each dimension to compute the individual elements on the diagonal, and then sum them up as follows

$$\mathrm{tr}\left(\boldsymbol{F}_t(\boldsymbol{x}_t, t)\right) \approx \frac{1}{\sigma_t} \sum_{i=1}^{d} \frac{\partial \left[\left\langle \boldsymbol{\varepsilon}_\theta(\boldsymbol{x}_t, t) \middle| \boldsymbol{e}^{(i)} \right\rangle\right]}{\partial \boldsymbol{x}_t}. \quad (15)$$

Evaluating the trace using the VJP method would be extremely time-consuming due to the curse of dimensionality. If the time-complexity of a single backpropagation is $\mathcal{O}(d)$, then the calculation in equation 15 would have a time-complexity of $\mathcal{O}(d^2)$. In practice, as demonstrated in Table 1, evaluating the trace of diffusion Fisher on the SD-v1.5 model would require half an hour, rendering it nearly infeasible. Some Monte-Carlo type approximations can be used to accelerate the NLL evaluation of models, but they still fall short in per-sample NLL evaluation, see a detailed discussion in C.5.

**Log-Likelihood Evaluation via DF trace matching**  To overcome the limitations of the VJP method in evaluating

| Methods | The relative error of NLL evaluation | | | | | |
|---|---|---|---|---|---|---|
| | t = 1.0 | t = 0.8 | t = 0.6 | t = 0.4 | t = 0.2 | t = 0.0 |
| VJP (eq. 15) | 6.68% | 5.79% | 10.46% | 20.13% | 51.14% | 70.95% |
| DF-TM (**Ours**) | **3.41%** | **4.56%** | **4.13%** | **4.28%** | **5.33%** | **5.81%** |

Table 2: Comparison of the VJP method and our DF-TM in terms of the diffusion Fisher trace evaluation error across different timesteps. The error is evaluated on the 2-D chessboard data with the VE schedule.

the trace of the diffusion Fisher, we propose to directly obtain its analytical form and train a network to match it. Note that the trace of an outer-product of the vector results in the vector's norm. Thus, given the diffusion Fisher in Proposition 1, we can also derive its trace in the form of a weighted norm sum, as highlighted in the following proposition:

*Proposition* 5. In the same context as Proposition 1, the trace of the diffusion Fisher matrix for the diffused distribution $q_t$, where $t \in (0, 1]$, is given by:

$$\mathrm{tr}\left(\boldsymbol{F}_t(\boldsymbol{x}_t, t)\right) = \frac{d}{\sigma_t^2} - \frac{\alpha_t^2}{\sigma_t^4} \left[\sum_i w_i \|\boldsymbol{y}_i\|^2 - \left\|\sum_i w_i \boldsymbol{y}_i\right\|^2\right] \quad (16)$$

As demonstrated in Proposition 2, the $\left\|\sum_i w_i \boldsymbol{y}_i\right\|^2$ can be directly estimated by $\|\bar{\boldsymbol{y}}_\theta(\boldsymbol{x}_t, t)\|^2$. Therefore, the only

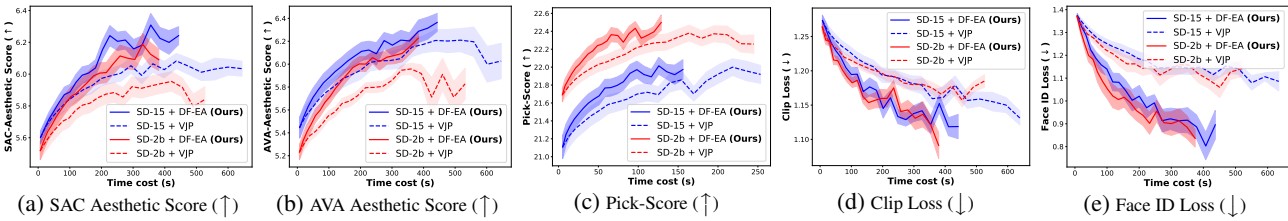

(a) SAC Aesthetic Score ($\uparrow$)  (b) AVA Aesthetic Score ($\uparrow$)  (c) Pick-Score ($\uparrow$)  (d) Clip Loss ($\downarrow$)  (e) Face ID Loss ($\downarrow$)

Figure 3: Comparison between our DF method and the VJP method on adjoint guidance sampling across five objective scores: SAC/AVA aesthetic score, Pick-Score, clip loss, and Face ID loss. Notably, our DF consistently achieves superior scores with less time expenditure.

unknown element in equation 16 is $\sum_i w_i \|\boldsymbol{y}_i\|^2$. Consequently, we suggest learning this term using a scalar-valued network, as per the following training scheme:

---

**Algorithm 1** Training of DF-TM Network

---

1: **Input**: data space dimension $d$, initial network $\boldsymbol{t}_\theta(\cdot, \cdot) : \mathbb{R}^d \times \mathbb{R} \mapsto \mathbb{R}$, noise schedule $\{\alpha_t\}$ and $\{\sigma_t\}$.
2: **repeat**
3: $\quad \boldsymbol{x}_0 \sim q_0(\boldsymbol{x}_0)$
4: $\quad t \sim \text{Uniform}(\{1, \dots, T\})$
5: $\quad \boldsymbol{\varepsilon} \sim \mathcal{N}(\mathbf{0}, \mathbf{I})$
6: $\quad \boldsymbol{x}_t = \alpha_t \boldsymbol{x}_0 + \sigma_t \boldsymbol{\varepsilon}$
7: $\quad$ Take gradient descent on $\nabla_\theta \left| \boldsymbol{t}_\theta(\boldsymbol{x}_t, t) - \frac{\|\boldsymbol{x}_0\|^2}{d} \right|^2$
8: **until** converged
9: **Output**: $\boldsymbol{t}_\theta(\cdot, \cdot)$

---

The training scheme detailed in Algorithm 1 can indeed enable $\boldsymbol{t}_\theta(\boldsymbol{x}_t, t)$ to estimate the weighted norm term $\frac{1}{d} \sum_i w_i(\boldsymbol{x}_t, t) \|\boldsymbol{y}_i\|^2$. This is substantiated by the convergence analysis Proposition 6, as presented below.

> *Proposition* 6. $\forall (x_t, t) \in \mathbb{R}^d \times \mathbb{R}_{\geq 0}$, the optimal $t_\theta(\boldsymbol{x}_t, t)$s trained by the objective in Algorithm 1 are equal to $\frac{1}{d} \sum_i w_i(\boldsymbol{x}_t, t) \|\boldsymbol{y}_i\|^2$.

Once we have obtained $\boldsymbol{t}_\theta$, we can evaluate the trace of diffusion Fisher efficiently, as illustrated below. This approach is a straightforward result of equation 16 and Propositions 2 and 6, which we refer to as DF trace matching (DF-TM).

$$\text{tr}\left(\boldsymbol{F}_t(\boldsymbol{x}_t, t)\right) \approx \frac{d}{\sigma_t^2} - \frac{\alpha_t^2}{\sigma_t^4}\left(d * t_\theta(\boldsymbol{x}_t, t) - \|\boldsymbol{y}_\theta(\boldsymbol{x}_t, t)\|^2\right) \tag{17}$$

To estimate the trace using DF-TM in equation 17, we simply need one access to $\boldsymbol{t}_\theta$ and $\boldsymbol{\varepsilon}_\theta$. DF-TM enables us to effectively evaluate the log-likelihood in a gradient-free manner with linear time complexity. We can further substantiate the theoretical approximation error bound when using DF-TM to calculate the trace of the diffusion Fisher, as illustrated in the Proposition 7.

> *Proposition* 7. Assume the approximation error on $t_\theta(\boldsymbol{x}_t, t)$ is $\delta_1$ and on $\varepsilon_\theta(\boldsymbol{x}_t, t)$ is $\delta_2$, then the approximation error of the approximated Fisher trace in equation 17 is at most $\frac{\alpha_t^2}{\sigma_t^4}\delta_1 + \frac{1}{\sigma_t^2}\delta_2^2$.

**Experiments** We conduct toy experiments to show the accuracy and commercial-level experiments to show the efficiency of our DF-TM method. In toy experiments, as shown in Table 2, we trained a simple DF-TM network on 2-D chessboard data and then evaluated the relative error of the trace estimated by VJP and DF-TM. It is shown that our DF-TM method consistently achieves more accurate trace estimation of DF compared to the VJP method.

In commercial-level experiments, we trained two DF-TM networks for the SD-1.5 and SD-2base pipeline on the Laion2B-en dataset (Schuhmann et al., 2022), which contains 2.32 billion text-image pairs. Our $t_\theta$ follows the U-net structure, similar to stable diffusion models, but with an added MLP head to produce a scalar-valued output. We utilize the AdamW optimizer (Loshchilov & Hutter, 2019) with a learning rate of 1e-4. The training is executed across 8 V100 chips with a batch size of 384 and completed after 150K steps. In Figure 1a, we demonstrate that the training loss of DF-TM nets converges smoothly, indicating the robustness of the DF-TM training scheme in Algorithm 1.

In Figure 1b, we evaluate the average NLL and Clip score of samples generated by various SD models, using 10k randomly selected prompts from the COCO dataset (Lin et al., 2014). A lower NLL suggests more realistic data generation, while a higher Clip score indicates a better match between the generated images and the input prompts. The results imply that the NLL and the Clip score form a trade-off curve across different guidance scales. This phenomenon, previously hypothesized in theory (Wu et al., 2024), is now confirmed in the SD models, thanks to the effective NLL evaluation via the DF-TM method.

In Figure 2, we display images with varying NLL under the same prompt with 10 steps on SD-1.5 using DDIM. It's clear that images with lower NLL exhibit greater visual realism, while those with higher NLL often contain deformed elements (emphasized by the yellow rectangle). Our proposed

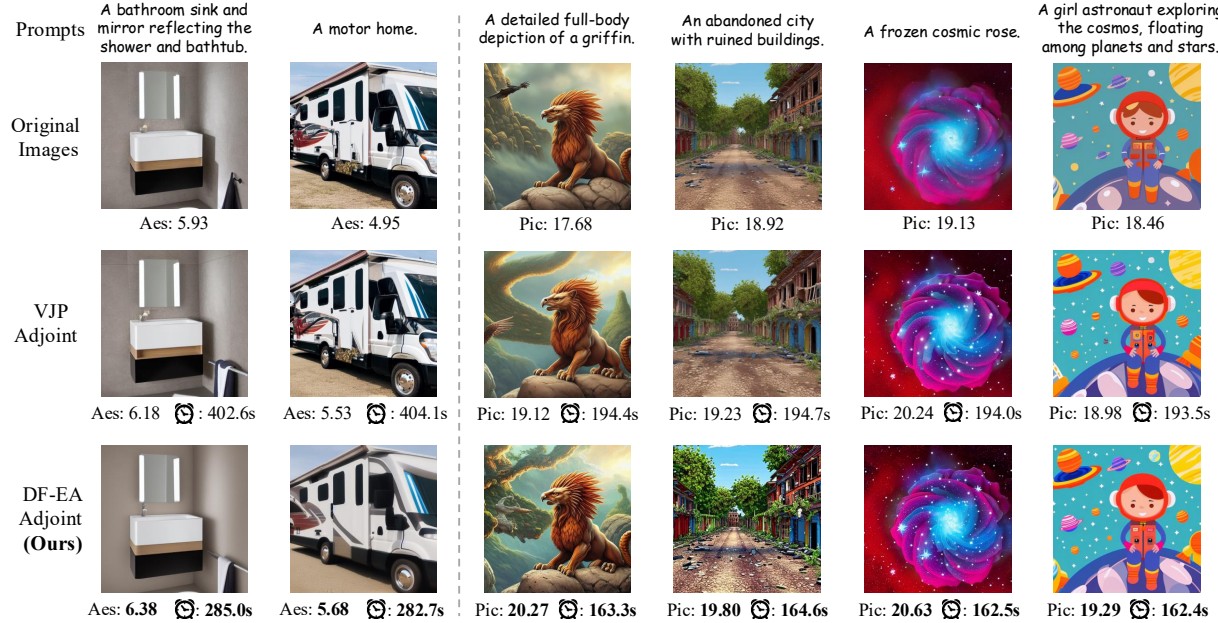

Figure 4: Visual comparison of DF-EA (**Ours**) and VJP in the adjoint improvement task on (left) SAC aesthetic score and (right) Pick-Score. DF-EA consistently generates images with better visual effects and reduced time expenditure.

NLL evaluation method proves to be an effective tool for automatic sample selection.

## 5. Diffusion Fisher Endpoint Approximation

The adjoint optimization of DMs would require access to the matrix-vector multiplication of the diffusion Fisher, which can also be seen as applying diffusion Fisher as a linear operator. In this section, we present a training-free method that simplifies the complex linear transformation calculations, thus enabling faster and more accurate adjoint optimization based on the outer-product form diffusion Fisher.

**Adjoint optimization sampling.** Guided sampling techniques are extensively utilized in diffusion models to facilitate controllable generation. Recently, to address the inflexibility of commonly used classifier-based guidance (Dhariwal & Nichol, 2021b) and classifier-free guidance (Ho & Salimans, 2022), a series of adjoint guidance methods have been explored (Pan et al., 2023a;b).

Consider optimizing a scalar-valued loss function $\mathcal{L}(\cdot) : \mathbb{R}^d \mapsto \mathbb{R}$, which takes $\boldsymbol{x}_0$ in the data space as input. Adjoint guidance is implemented by applying gradient descent on $\boldsymbol{x}_t$ in the direction of $\frac{\partial \mathcal{L}(\boldsymbol{x}_0(\boldsymbol{x}_t))}{\partial \boldsymbol{x}_t}$. The essence of adjoint guidance is to use the gradient at $t = 0$ and follow the adjoint ODE (Pollini et al., 2018; Chen et al., 2018) to compute $\boldsymbol{\lambda}_t := \frac{\partial \mathcal{L}(\boldsymbol{x}_0(\boldsymbol{x}_t))}{\partial \boldsymbol{x}_t}$ for $t > 0$.

$$\frac{\mathrm{d}\boldsymbol{\lambda}_t}{\mathrm{d}t} = -\boldsymbol{\lambda}_t^\top \frac{\partial \boldsymbol{h}_\theta(\boldsymbol{x}_t, t)}{\partial \boldsymbol{x}_t}, \quad \boldsymbol{\lambda}_0 = \frac{\partial \mathcal{L}(\boldsymbol{x}_0)}{\partial \boldsymbol{x}_0} \quad (18)$$

**Adjoint ODE via VJP.** Regardless of the ODE solver being used, it is necessary to compute the right-hand side of equation 18, or equivalently, $\boldsymbol{F}(\boldsymbol{x}_t, t)^\top \boldsymbol{\lambda}_t$. This computation can be interpreted as applying the diffusion Fisher matrix as a linear operator to the adjoint state $\boldsymbol{\lambda}_t$, from a functional analysis perspective (Yosida, 2012). Current practices utilize the VJP technique to approximate this linear transformation operation as follows:

$$\begin{aligned}
\boldsymbol{F}(\boldsymbol{x}_t, t)^\top \boldsymbol{\lambda}_t &\approx \frac{1}{\sigma_t} \frac{\partial \boldsymbol{\varepsilon}_\theta(\boldsymbol{x}_t, t)^\top}{\partial \boldsymbol{x}_t} \boldsymbol{\lambda}_t \\
&\approx \frac{1}{\sigma_t} \frac{\partial \left[\langle \boldsymbol{\varepsilon}_\theta(\boldsymbol{x}_t, t) | \boldsymbol{\lambda}_t \rangle\right]}{\partial \boldsymbol{x}_t}
\end{aligned} \quad (19)$$

This process involves computationally expensive neural network auto-differentiations, and the approximation errors introduced by the VJP technique have no theoretical bound.

**Adjoint ODE via DF-EA.** As previously discussed in Section 3, the DF inherently doesn't require gradients, suggesting that we could potentially apply the diffusion Fisher as a linear operator in a gradient-free manner. The challenging part in equation 11 is $\sum_i w_i \boldsymbol{y}_i \boldsymbol{y}_i^\top$, which represents a weighted form of outer-products of data. Based on the definition of $w_i$, the closest $\boldsymbol{y}_i$ to $\boldsymbol{x}_0$ will dominate as $t \to 0$. This makes it intuitive to replace this sum with a single final sample outer-product $\boldsymbol{x}_0 \boldsymbol{x}_0^\top$. It's also important to note that the adjoint guidance itself needs to compute $\boldsymbol{x}_0$ at each guidance step, eliminating the need for additional computation to obtain $\boldsymbol{x}_0$. Given that we utilize the endpoint sample $\boldsymbol{x}_0$, we refer to this approximation technique as DF Endpoint

Approximation (EA). The formulation for DF-EA in the adjoint ODE is as follows:

$$
\begin{aligned}
&\boldsymbol{F}(\boldsymbol{x}_t, t)^{\top} \boldsymbol{\lambda}_t \\
&\approx \left( \frac{1}{\sigma_t^2} \boldsymbol{I} - \frac{\alpha_t^2}{\sigma_t^4} \left( \sum_i w_i \boldsymbol{y}_i \boldsymbol{y}_i^{\top} - \bar{\boldsymbol{y}}_\theta(\boldsymbol{x}_t, t) \bar{\boldsymbol{y}}_\theta(\boldsymbol{x}_t, t)^{\top} \right) \right)^{\top} \boldsymbol{\lambda}_t \\
&\approx \left( \frac{1}{\sigma_t^2} \boldsymbol{I} - \frac{\alpha_t^2}{\sigma_t^4} \left( \boldsymbol{x}_0 \boldsymbol{x}_0^{\top} - \bar{\boldsymbol{y}}_\theta(\boldsymbol{x}_t, t) \bar{\boldsymbol{y}}_\theta(\boldsymbol{x}_t, t)^{\top} \right) \right)^{\top} \boldsymbol{\lambda}_t \\
&= \frac{1}{\sigma_t^2} \boldsymbol{\lambda}_t - \frac{\alpha_t^2}{\sigma_t^4} \langle \boldsymbol{x}_0, \boldsymbol{\lambda}_t \rangle \boldsymbol{x}_0 + \frac{\alpha_t^2}{\sigma_t^4} \langle \bar{\boldsymbol{y}}_\theta(\boldsymbol{x}_t, t), \boldsymbol{\lambda}_t \rangle \bar{\boldsymbol{y}}_\theta(\boldsymbol{x}_t, t)
\end{aligned}
$$

(20)

The DF-EA approximation leads to a scalar-weighted combination of $\boldsymbol{\lambda}_t$, $\boldsymbol{x}_0$, and $\bar{\boldsymbol{y}}_\theta(\boldsymbol{x}_t, t)$, which importantly does not involve any gradients. Additionally, we derive the theoretical approximation error bound of the DF-EA in Proposition 8. To measure the accuracy of DF-EA as a linear operator, we opt to use the Hilbert–Schmidt norm (Gohberg et al., 1990) for measurement, as follows:

*Proposition* 8. Assume that the approximation error on $\boldsymbol{\varepsilon}_\theta(\boldsymbol{x}_t, t)$ is $\delta_2$, the approximation error of the DF-EA linear operator, as referenced in 20, is at most $\frac{\alpha_t^2}{\sigma_t^3} \left( 2\mathcal{D}_y^2 + \sqrt{d}\delta_2 \right)$ when measured in terms of the Hilbert–Schmidt norm.

**Experiments on DF-EA.** As depicted in Figure 3, we conducted experiments comparing our DF-EA and VJP methods in adjoint guidance sampling, using five different scores and two different base models. DF-EA consistently achieves better scores due to its bounded approximation error. Furthermore, DF-EA requires less processing time as it eliminates the need for time-consuming gradient operations. DF-EA and VJP are compared under the same guidance scales and schemes across various numbers of steps. Details regarding the score function can be found in Appendix B.2.

As depicted in Figure 4, our DF-EA consistently generates samples with higher scores with a reduced time cost compared to VJP. All samples are generated within 50 steps, with adjoint applied from the 15[th] to the 35[th] step. Details on hyperparameters can be found in Appendix B.2.

## 6. Numerical OT Verification of DMs

There is an increasing trend towards analyzing the probability modeling capabilities of DMs by interpreting them from an optimal transport (OT) perspective (Albergo et al., 2023; Chen et al., 2024). The foundational concepts of optimal transport can be found in Appendix A.9. One of the central questions is whether the map deduced by the PF-ODE could represent an optimal transport. Khrulkov et al. (2023) have proven that, given single-Gaussian initial data and a VE noise schedule, the PF-ODE deduced map is

optimal transport. Zhang et al. (2024a) has demonstrated that affine initial data suffices for the PF-ODE map to be optimal transport. However, the OT property of the general PF-ODE map remains an open question.

In this section, we propose the first numerical OT verification experiment for a general PF-ODE deduced map based on the outer-product form diffusion Fisher we obtained in section 3. We first derive the following corollary for the OT property of the PF-ODE deduced map.

*Corollary* 1. Denote the diffeomorphism deduced by the PF-ODE in equation 5 as follows

$$
T_{s,t} : \mathbb{R}^n \longrightarrow \mathbb{R}^n; \boldsymbol{x}_s \longmapsto \boldsymbol{x}_t, \quad \forall t \geq s > 0. \quad (21)
$$

The diffeomorphism $T_{s,T}$ is a Monge optimal transport map **if and only if** the normalized fundamental matrix for $\boldsymbol{B}(t) \equiv \boldsymbol{B}(t, \boldsymbol{x}_t)$ at $s$ is s.p.d. for every PF-ODE chain that starts from a $\boldsymbol{x}_T \in \mathbb{R}^d$. where

$$
\begin{aligned}
\boldsymbol{B}(t, \boldsymbol{x}_t) = {} & \left[ f(t) - \frac{g^2(t)}{2\sigma_t^2} \right] \boldsymbol{I} + \frac{\alpha_t^2 g^2(t)}{2\sigma_t^4} \Bigg[ \sum_i w_i \boldsymbol{y}_i \boldsymbol{y}_i^{\top} \\
& - \left( \sum_i w_i \boldsymbol{y}_i \right) \left( \sum_i w_i \boldsymbol{y}_i \right)^{\top} \Bigg].
\end{aligned}
$$

(22)

The definition of the normalized fundamental matrix is deferred to Appendix A.10.

Note that $T_{s,t}$ is well-posed, guaranteed by the global version of the Picard-Lindelöf theorem (Amann, 2011; Zhang et al., 2024a). Detailed proofs can be found in Appendix A.10. We then design the numerical OT verification experiment for a given noise schedule and initial data as shown in the Algorithm 2. The detailed version of Algorithm 2 can be found in Appendix B.3.

---

**Algorithm 2** Numerical OT test for PF-ODE map

1: **Input**: initial data $\{\boldsymbol{y}_i\}_{i=1}^N$, noise schedule $\{\alpha_t\}$ and $\{\sigma_t\}$, discretization steps $M$.
2: Initialize $\boldsymbol{A}_M = \boldsymbol{I}$, $\boldsymbol{x}_M \sim \mathcal{N}(0, \sigma_T \boldsymbol{I})$.
3: **for** $i = M, M-1, \cdots, 1$ **do**
4: $\quad$ $\mathrm{d}t = t_{i-1} - t_i$.
5: $\quad$ Calculate $\boldsymbol{B}_i$ by equation 22.
6: $\quad$ $\boldsymbol{A}_{i-1} = \boldsymbol{A}_i + \mathrm{d}t * \boldsymbol{A}_i^{\top} \boldsymbol{B}_i$ $\quad$ {solve fundamental matrix.}
7: $\quad$ $\boldsymbol{x}_{i-1} = \text{PF-ODE Solver}(\boldsymbol{x}_i, i)$
8: **end for**
9: **Output**: $\boldsymbol{A}_0$. $\qquad$ {The result fundamental matrix.}

---

In Table 3, we numerically tested the OT property of the PF-ODE map under different nose schedules and initial data. We tested four commonly used noise schedules, VE (Ho et al., 2020), VP (Song & Ermon, 2019), sub-VP (Song et al., 2020), and EDM (Karras et al., 2022) on 2-D data. We first tested when the initial data is single-Gaussian and affine, we found that the result normalized fundamental

| Initial Data | Single-Gaussian | | Affine | | Non-affine | |
|---|---|---|---|---|---|---|
| Noise Schedule | Asym. | OT | Asym. | OT | Asym. | OT |
| VE (Song & Ermon, 2019) | 0.00% | ✓ | 0.00% | ✓ | 25.28% | ✗ |
| VP (Ho et al., 2020) | 0.00% | ✓ | 0.00% | ✓ | 23.36% | ✗ |
| sub-VP (Song et al., 2020) | 0.00% | ✓ | 0.00% | ✓ | 13.84% | ✗ |
| EDM (Karras et al., 2022) | 0.00% | ✓ | 0.00% | ✓ | 27.09% | ✗ |

Table 3: Comparison of numerical OT verification results of four commonly used noise schedulers with different initial data.

matrix is p.s.d., which means the PF-ODE map is optimal transport. These results coincide with the previous proposed theoretical results in (Khrulkov et al., 2023) and (Zhang et al., 2024a). We further tested an extremely simple non-affine case, that is, there are only three initial data points, (0,0), (0,0.5), and (0.5,0). We showed that all noise schedules result in an obvious asymmetric fundamental matrix, which means that the PF-ODE map fails to be OT in these cases. Here, we use the Frobenius norm of the difference of $A$ and its transpose divided by the Frobenius norm of $A$ to measure its asymmetric rate. Based on this finding, we hypothesize that the PF-ODE map can only be OT in the affine initial data (Single-Gaussian is a special case of affine data), but fails to be OT in general non-affine data. Our hypothesis coincides with the statement that the flow map of a Fokker–Planck equation is not necessarily OT as proposed in (Lavenant & Santambrogio, 2022). (Note that PF-ODE is a special class of Fokker–Planck equations)

## 7. Conclusions

This paper derives an outer-product span formulation of the diffusion Fisher and introduces two approximation algorithms for different types of access to the diffusion Fisher: diffusion Fisher trace matching (DF-TM) and diffusion Fisher endpoint approximation (DF-EA). Both methods are theoretically guaranteed in terms of approximation error bounds, and offer improved accuracy and reduced time-cost compared to the traditional VJP method. We also designed the numerical OT property verification experiment for the PF-ODE map based on the outer-product form of DF. This work not only improves the efficiency of diffusion Fisher access but also widens our understanding of the diffusion models. Please refer to further discussions in appendix C. The code is available at `https://github.com/zituitui/DiffusionFisher`.

## Acknowledgments

This work was supported in part by the National Natural Science Foundation of China under Grants 62206248 and 62402430, and the Zhejiang Provincial Natural Science Foundation of China under Grant LQN25F020008. We would like to thank all the reviewers for their constructive comments. Fangyikang Wang wishes to express gratitude to Pengze Zhang from ByteDance, Binxin Yang, and Xinhang Leng from WeChat Vision for their insightful discussions on the experiments.

## Impact Statement

The development of accurate diffusion Fisher, as discussed in this paper, holds significant potential for several domains, including machine learning, healthcare, environmental modeling, and economics. However, while this research holds great potential for positive impacts, it is also important to consider potential negative societal impacts. The enhanced ability of generative models given by DF could potentially be misused. For instance, it could be exploited to create aesthetically improved deepfakes, leading to misinformation. In healthcare, if not properly regulated, the use of synthetic patient data could lead to ethical issues. Therefore, it is crucial to ensure that the findings of this research are applied ethically and responsibly, with necessary safeguards in place to prevent misuse and protect privacy.

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

# Appendix

## A. Proofs and Formulations

### A.1. Proof of Proposition 1

Notice that, in the subsection, we can do an interchange of sum and gradient; this is due to Leibniz's rule (Osler, 1970) and the boundness condition we set in equation 10. Before we give the proof of Proposition 1, we would like to establish two technical lemmas. The first lemma is about the first partial derivative of $v_i(\boldsymbol{x}_t, t)$ with respect to $\boldsymbol{x}_t$.

*Lemma* 1.

$$
\begin{aligned}
\frac{\partial v_i(\boldsymbol{x}_t, t)}{\partial \boldsymbol{x}_t} &= \frac{\partial \exp\left(-\frac{|\boldsymbol{x}_t - \alpha_t \boldsymbol{y}_i|^2}{2\sigma_t^2}\right)}{\partial \boldsymbol{x}_t} \\
&= -\frac{1}{\sigma_t^2}(\boldsymbol{x}_t - \alpha_t \boldsymbol{y}_i) \exp\left(-\frac{|\boldsymbol{x}_t - \alpha_t \boldsymbol{y}_i|^2}{2\sigma_t^2}\right) \\
&= -\frac{1}{\sigma_t^2}(\boldsymbol{x}_t - \alpha_t \boldsymbol{y}_i) v_i(\boldsymbol{x}_t, t)
\end{aligned}
\tag{23}
$$

The second lemma is about the Jacobian of $\sum_i w_i(\boldsymbol{x}_t, t)\, \boldsymbol{y}_i$ w.r.t. $\boldsymbol{x}_t$.

*Lemma* 2.

$$
\begin{aligned}
& \frac{\partial \sum_i w_i(\boldsymbol{x}_t, t)\, \boldsymbol{y}_i}{\partial \boldsymbol{x}_t} \\
&= \sum_i \boldsymbol{y}_i \left(\frac{\partial w_i(\boldsymbol{x}_t, t)}{\partial \boldsymbol{x}_t}\right)^\top \\
&= \sum_i \boldsymbol{y}_i \left(\frac{\partial}{\partial \boldsymbol{x}_t}\left[\frac{v_i(\boldsymbol{x}_t, t)}{\sum_j v_j(\boldsymbol{x}_t, t)}\right]\right)^\top \\
&= \sum_i \boldsymbol{y}_i \left\{\frac{\frac{\partial v_i(\boldsymbol{x}_t, t)}{\partial \boldsymbol{x}_t}[\sum_k v_k(\boldsymbol{x}_t, t)] - v_i(\boldsymbol{x}_t, t)\frac{\partial}{\partial \boldsymbol{x}_t}\left[\sum_j v_j(\boldsymbol{x}_t, t)\right]}{[\sum_k v_k(\boldsymbol{x}_t, t)]^2}\right\}^\top \\
&= \frac{1}{(\sum_k v_k)^2}\sum_i \boldsymbol{y}_i \left\{-\frac{1}{\sigma_t^2}(\boldsymbol{x}_t - \alpha_t \boldsymbol{y}_i) v_i \sum_k v_k - v_i(\boldsymbol{x}_t, t)\left(-\frac{1}{\sigma_t^2}\right)\sum_j (\boldsymbol{x}_t - \alpha_t \boldsymbol{y}_j) v_j\right\}^\top \\
&= \frac{1}{(\sum_k v_k)^2}\left(-\frac{1}{\sigma_t^2}\right)\sum_i v_i \boldsymbol{y}_i \left\{(\boldsymbol{x}_t - \alpha_t \boldsymbol{y}_i)\sum_k v_k - \sum_j (\boldsymbol{x}_t - \alpha_t \boldsymbol{y}_i) v_j\right\}^\top \\
&= \frac{1}{(\sum_k v_k)^2}\left(-\frac{1}{\sigma_t^2}\right)\sum_i v_i \boldsymbol{y}_i \left\{-\alpha_t \boldsymbol{y}_i \sum_k v_k + \alpha_t \sum_j \boldsymbol{y}_j v_j\right\}^\top \\
&= \frac{\alpha_t}{\sigma_t^2}\left[\frac{\sum_i v_i \boldsymbol{y}_i \boldsymbol{y}_i^\top(\sum_k v_k)}{(\sum_k v_k)^2} - \left(\frac{\sum_i v_i \boldsymbol{y}_i}{\sum_k v_k}\right)\left(\frac{\sum_i v_i \boldsymbol{y}_i}{\sum_k v_k}\right)^\top\right] \\
&= \frac{\alpha_t}{\sigma_t^2}\left[\sum_i w_i \boldsymbol{y}_i \boldsymbol{y}_i^\top - \left(\sum_i w_i \boldsymbol{y}_i\right)\left(\sum_i w_i \boldsymbol{y}_i\right)^\top\right]
\end{aligned}
\tag{24}
$$

Now we are ready to give the Proof of Proposition 1

*Proof.* According to the initial distribution (equation 10) and the diffusion kernel (equation 1), the marginal distribution at

some time $t > 0$ would be

$$p\left(\boldsymbol{x}_t, t\right) = \frac{1}{N} \sum_i \left(2\pi\sigma_t^2\right)^{-\frac{d}{2}} \exp\left(-\frac{|x_t - \alpha_t \boldsymbol{y}_i|^2}{2\sigma_{t^2}}\right) \tag{25}$$

Thus, the log-density has the following analytical formulation

$$\begin{aligned}
\log p(\boldsymbol{x}_t, t) &= \log\left[\frac{1}{N}\left(2\pi\sigma_t^2\right)^{-\frac{d}{2}} \sum_i \exp\left(-\frac{|x_t - \alpha_t \boldsymbol{y}_i|^2}{2\sigma_{t^2}}\right)\right] \\
&= \log\left[\sum_i \exp\left(-\frac{|x_t - \alpha_t \boldsymbol{y}_i|^2}{2\sigma_t^2}\right)\right] + C \\
&= \log\left[\sum_i v_i\left(\boldsymbol{x}_t, t\right)\right] + C
\end{aligned} \tag{26}$$

The score can be expressed as follows

$$\begin{aligned}
\frac{\partial}{\partial \boldsymbol{x}_t} \log p\left(\boldsymbol{x}_t, t\right) &= \frac{\partial}{\partial \boldsymbol{x}_t} \log\left[\sum_i v_i\left(\boldsymbol{x}_t, t\right)\right] \\
&= \frac{\frac{\partial}{\partial \boldsymbol{x}_t}\left[\sum_i v_i\left(\boldsymbol{x}_t, t\right)\right]}{\sum_i v_i\left(\boldsymbol{x}_t, t\right)} \\
&= \frac{-\frac{1}{\sigma_t^2} \sum_j \left(\boldsymbol{x}_t - \alpha_t y_j\right) v_j}{\sum_i v_i\left(\boldsymbol{x}_t, t\right)} \\
&= -\frac{1}{\sigma_t^2}\left[x_t - \alpha_t \sum_j w_j\left(\boldsymbol{x}_t, t\right) y_j\right]
\end{aligned} \tag{27}$$

The Fisher information we want can then be calculated by further applying a gradient on the score.

$$\begin{aligned}
\boldsymbol{F}_t(\boldsymbol{x}_t, t) &= -\frac{\partial}{\partial \boldsymbol{x}_t}\left(\frac{\partial}{\partial \boldsymbol{x}_t} \log p\left(\boldsymbol{x}_t, t\right)\right) \\
&= -\frac{\partial}{\partial \boldsymbol{x}_t}\left\{-\frac{1}{\sigma_t^2}\left[x_t - \alpha_t \sum_j w_j\left(\boldsymbol{x}_t, t\right) y_j\right]\right\} \\
&= \frac{1}{\sigma_t^2}\boldsymbol{I} - \frac{\alpha_t}{\sigma_t^2}\frac{\partial \sum_i w_i\left(\boldsymbol{x}_t, t\right) \boldsymbol{y}_i}{\partial \boldsymbol{x}_t} \\
&= \frac{1}{\sigma_t^2}\boldsymbol{I} - \frac{\alpha_t^2}{\sigma_t^4}\left[\sum_i w_i \boldsymbol{y}_i \boldsymbol{y}_i^\top - \left(\sum_i w_i \boldsymbol{y}_i\right)\left(\sum_i w_i \boldsymbol{y}_i\right)^\top\right] \quad \text{(by Lemma 2)}
\end{aligned} \tag{28}$$

$\square$

This proof is inherently the calculation of the Hessian of a log-convolution of a density, we provide the detailed derivation here for completeness.

**A.2. Proof of Proposition 2**

*Proof.* Given fixed $(x_t, t)$, $\mathcal{L}$ is a quadratic form of $y_\theta$. To obtain the optimal $\bar{\boldsymbol{y}}_\theta$, we differentiate $\mathcal{L}$ and set this derivative equal to zero, resulting in the following

$$0 = \frac{\partial \mathcal{L}}{\partial \bar{\boldsymbol{y}}_\theta(\boldsymbol{x}_t, t)} = \frac{\partial}{\partial \bar{\boldsymbol{y}}_\theta(\boldsymbol{x}_t, t)} \sum_j \underbrace{\frac{1}{N}(2\pi\sigma_t^2)^{-\frac{d}{2}} v_j(\boldsymbol{x}_t, t)\lambda_t}_{A_t} \frac{\alpha_t^2}{\sigma_t^2} \|\bar{\boldsymbol{y}}_\theta(\boldsymbol{x}_t, t) - \boldsymbol{y}_j\|^2$$

$$= 2A_t\lambda_t\frac{\alpha_t^2}{\sigma_t^2} \sum_j \boldsymbol{v}_j(x_t, t)(\bar{\boldsymbol{y}}_\theta(\boldsymbol{x}_t, t) - \boldsymbol{y}_j), \tag{29}$$

which yields

$$\bar{\boldsymbol{y}}_\theta^*(\boldsymbol{x}_t, t) = \sum_k \frac{v_k(\boldsymbol{x}_t, t)}{\sum_j v_j(\boldsymbol{x}_t, t)}\boldsymbol{y}_k = \sum_i w_i\boldsymbol{y}_i. \tag{30}$$

$\square$

## A.3. Proof of Proposition 3

Notice that, in the subsection, we can do an interchange of integral and gradient; this is due to Leibniz's rule (Osler, 1970) and the bounded moments condition we set in equation 12. Before we give the proof of Proposition 3, we would like to establish two technical lemmas. The first lemma is about the first partial derivative of $v(\boldsymbol{x}_t, t, \boldsymbol{y})$ w.r.t. $\boldsymbol{x}_t$.

*Lemma* 3.

$$\begin{aligned}\frac{\partial v(\boldsymbol{x}_t, t, \boldsymbol{y})}{\partial \boldsymbol{x}_t} &= \frac{\partial \exp\left(-\frac{|\boldsymbol{x}_t - \alpha_t\boldsymbol{y}|^2}{2\sigma_t^2}\right)}{\partial \boldsymbol{x}_t} \\ &= -\frac{1}{\sigma_t^2}(\boldsymbol{x}_t - \alpha_t\boldsymbol{y})\exp\left(-\frac{|\boldsymbol{x}_t - \alpha_t\boldsymbol{y}|^2}{2\sigma_t^2}\right) \\ &= -\frac{1}{\sigma_t^2}(\boldsymbol{x}_t - \alpha_t\boldsymbol{y})v(\boldsymbol{x}_t, t, \boldsymbol{y})\end{aligned} \tag{31}$$

*Lemma* 4.

$$\begin{aligned}&\frac{\partial \int_{\mathbb{R}^d} w(\boldsymbol{x}_t, t, \boldsymbol{y})\boldsymbol{y}\mathrm{d}q_0(\boldsymbol{y})}{\partial \boldsymbol{x}_t} \\ &= \int_{\mathbb{R}^d} \boldsymbol{y}\left(\frac{\partial w(\boldsymbol{x}_t, t, \boldsymbol{y})}{\partial \boldsymbol{x}_t}\right)^\top \mathrm{d}q_0(\boldsymbol{y}) \\ &= \int_{\mathbb{R}^d} \boldsymbol{y}\left(\frac{\partial}{\partial \boldsymbol{x}_t}\left[\frac{v(\boldsymbol{x}_t, t, \boldsymbol{y})}{\int_{\mathbb{R}^d} v(\boldsymbol{x}_t, t, \boldsymbol{y}')\mathrm{d}q_0(\boldsymbol{y}')}\right]\right)^\top \mathrm{d}q_0(\boldsymbol{y}) \\ &= \int_{\mathbb{R}^d} \boldsymbol{y}\left\{\frac{\frac{\partial v_i(\boldsymbol{x}_t, t)}{\partial \boldsymbol{x}_t}\left[\int_{\mathbb{R}^d} v(\boldsymbol{x}_t, t, \boldsymbol{y}')\mathrm{d}q_0(\boldsymbol{y}')\right] - v(\boldsymbol{x}_t, t, \boldsymbol{y})\frac{\partial}{\partial \boldsymbol{x}_t}\left[\int_{\mathbb{R}^d} v(\boldsymbol{x}_t, t, \boldsymbol{y}')\mathrm{d}q_0(\boldsymbol{y}')\right]}{\left[\int_{\mathbb{R}^d} v(\boldsymbol{x}_t, t, \boldsymbol{y}'')\mathrm{d}q_0(\boldsymbol{y}'')\right]^2}\right\}^\top \mathrm{d}q_0(\boldsymbol{y}) \\ &= \frac{1}{\left[\int v(\boldsymbol{y}'')\mathrm{d}q_0(\boldsymbol{y}'')\right]^2}\int \boldsymbol{y}\left\{-\frac{(\boldsymbol{x}_t - \alpha_t\boldsymbol{y})v(\boldsymbol{y})}{\sigma_t^2}\int v(\boldsymbol{y}')\mathrm{d}q_0(\boldsymbol{y}') - v(\boldsymbol{y})\left(-\frac{1}{\sigma_t^2}\right)\int (\boldsymbol{x}_t - \alpha_t\boldsymbol{y}')v(\boldsymbol{y}')\mathrm{d}q_0(\boldsymbol{y}')\right\}^\top \mathrm{d}q_0(\boldsymbol{y}) \\ &= \frac{1}{\left[\int v(\boldsymbol{y}'')\mathrm{d}q_0(\boldsymbol{y}'')\right]^2}\left(-\frac{1}{\sigma_t^2}\right)\int v(\boldsymbol{y})\boldsymbol{y}\left\{(\boldsymbol{x}_t - \alpha_t\boldsymbol{y})\int v(\boldsymbol{y}')\mathrm{d}q_0(\boldsymbol{y}') - \int (\boldsymbol{x}_t - \alpha_t\boldsymbol{y}')v(\boldsymbol{y}')\mathrm{d}\boldsymbol{y}'\right\}^\top \mathrm{d}q_0(\boldsymbol{y}) \\ &= \frac{1}{\left[\int v(\boldsymbol{y}'')\mathrm{d}q_0(\boldsymbol{y}'')\right]^2}\left(-\frac{1}{\sigma_t^2}\right)\int v(\boldsymbol{y})\boldsymbol{y}\left\{-\alpha_t\boldsymbol{y}\int v(\boldsymbol{y}')\mathrm{d}q_0(\boldsymbol{y}') + \alpha_t\int v(\boldsymbol{y}')\boldsymbol{y}'\mathrm{d}\boldsymbol{y}'\right\}^\top \mathrm{d}q_0(\boldsymbol{y}) \\ &= \frac{\alpha_t}{\sigma_t^2}\left[\frac{\int v(\boldsymbol{y})\boldsymbol{y}\boldsymbol{y}^\top\mathrm{d}q_0(\boldsymbol{y}')\left[\int v(\boldsymbol{y}')\mathrm{d}q_0(\boldsymbol{y}')\right]}{\left[\int v(\boldsymbol{y}'')\mathrm{d}q_0(\boldsymbol{y}'')\right]^2} - \left(\frac{\int v(\boldsymbol{y})\boldsymbol{y}\mathrm{d}q_0(\boldsymbol{y})}{\int v(\boldsymbol{y}'')\mathrm{d}q_0(\boldsymbol{y}'')}\right)\left(\frac{\int v(\boldsymbol{y}')\boldsymbol{y}'\mathrm{d}q_0(\boldsymbol{y}')}{\int v(\boldsymbol{y}'')\mathrm{d}q_0(\boldsymbol{y}'')}\right)^\top\right] \\ &= \frac{\alpha_t}{\sigma_t^2}\left[\int w(\boldsymbol{y})\boldsymbol{y}\boldsymbol{y}^\top\mathrm{d}q_0(\boldsymbol{y}) - \left(\int w(\boldsymbol{y})\boldsymbol{y}\mathrm{d}q_0(\boldsymbol{y})\right)\left(\int w(\boldsymbol{y})\boldsymbol{y}\mathrm{d}q_0(\boldsymbol{y})\right)^\top\right]\end{aligned} \tag{32}$$

*Proof.* According to the initial distribution (equation 12) and the diffusion kernel (equation 1), the marginal distribution at

some time $t > 0$ would be

$$p\left(\boldsymbol{x}_t, t\right) = \int_{\mathbb{R}^d} \left(2\pi\sigma_t^2\right)^{-\frac{d}{2}} \exp\left(-\frac{|x_t - \alpha_t \boldsymbol{y}|^2}{2\sigma_{t^2}}\right) \mathrm{d}q_0(\boldsymbol{y}) \tag{33}$$

Thus the log-density has the following analytical formulation

$$
\begin{aligned}
\log q_t(\boldsymbol{x}_t, t) &= \log\left[\int_{\mathbb{R}^d} \left(2\pi\sigma_t^2\right)^{-\frac{d}{2}} \exp\left(-\frac{|x_t - \alpha_t \boldsymbol{y}|^2}{2\sigma_{t^2}}\right) \mathrm{d}q_0(\boldsymbol{y})\right] \\
&= \log\left[\int_{\mathbb{R}^d} \exp\left(-\frac{|x_t - \alpha_t \boldsymbol{y}|^2}{2\sigma_{t^2}}\right) \mathrm{d}q_0(\boldsymbol{y})\right] + C \\
&= \log\left[\int_{\mathbb{R}^d} v\left(\boldsymbol{x}_t, t, \boldsymbol{y}\right) \mathrm{d}q_0(\boldsymbol{y})\right] + C
\end{aligned}
\tag{34}
$$

The score can be expressed as follows

$$
\begin{aligned}
\frac{\partial}{\partial \boldsymbol{x}_t} \log p\left(\boldsymbol{x}_t, t\right) &= \frac{\partial}{\partial \boldsymbol{x}_t} \log\left[\int_{\mathbb{R}^d} v\left(\boldsymbol{x}_t, t, \boldsymbol{y}\right) \mathrm{d}q_0(\boldsymbol{y})\right] \\
&= \frac{\frac{\partial}{\partial \boldsymbol{x}_t}\left[\int_{\mathbb{R}^d} v\left(\boldsymbol{x}_t, t, \boldsymbol{y}\right) \mathrm{d}q_0(\boldsymbol{y})\right]}{\int_{\mathbb{R}^d} v\left(\boldsymbol{x}_t, t, \boldsymbol{y}\right) \mathrm{d}q_0(\boldsymbol{y})} \\
&= \frac{\int_{\mathbb{R}^d} \frac{\partial}{\partial \boldsymbol{x}_t}\left[v\left(\boldsymbol{x}_t, t, \boldsymbol{y}\right)\right] \mathrm{d}q_0(\boldsymbol{y})}{\int_{\mathbb{R}^d} v\left(\boldsymbol{x}_t, t, \boldsymbol{y}\right) \mathrm{d}q_0(\boldsymbol{y})} \quad \text{(by xx and Leibniz integral rule)} \\
&= \frac{-\frac{1}{\sigma_t^2}\int_{\mathbb{R}^d}(\boldsymbol{x}_t - \alpha_t \boldsymbol{y})v\left(\boldsymbol{x}_t, t, \boldsymbol{y}\right) \mathrm{d}q_0(\boldsymbol{y})}{\int_{\mathbb{R}^d} v\left(\boldsymbol{x}_t, t, \boldsymbol{y}\right) \mathrm{d}q_0(\boldsymbol{y})} \\
&= -\frac{1}{\sigma_t^2}\left[x_t - \alpha_t \int_{\mathbb{R}^d} w(\boldsymbol{x}_t, t, \boldsymbol{y})\boldsymbol{y}\mathrm{d}q_0(\boldsymbol{y})\right]
\end{aligned}
\tag{35}
$$

The Fisher information we want can then be calculated by further applying a gradient on the score.

$$
\begin{aligned}
\boldsymbol{F}_t(\boldsymbol{x}_t, t) &= -\frac{\partial}{\partial \boldsymbol{x}_t}\left(\frac{\partial}{\partial \boldsymbol{x}_t} \log p\left(\boldsymbol{x}_t, t\right)\right) \\
&= -\frac{\partial}{\partial \boldsymbol{x}_t}\left\{-\frac{1}{\sigma_t^2}\left[x_t - \alpha_t \int_{\mathbb{R}^d} w(\boldsymbol{x}_t, t, \boldsymbol{y})\boldsymbol{y}\mathrm{d}q_0(\boldsymbol{y})\right]\right\} \\
&= \frac{1}{\sigma_t^2}\boldsymbol{I} - \frac{\alpha_t}{\sigma_t^2}\frac{\partial \int_{\mathbb{R}^d} w(\boldsymbol{x}_t, t, \boldsymbol{y})\boldsymbol{y}\mathrm{d}q_0(\boldsymbol{y})}{\partial \boldsymbol{x}_t} \\
&= \frac{1}{\sigma_t^2}\boldsymbol{I} - \frac{\alpha_t^2}{\sigma_t^4}\left[\int w_i \boldsymbol{y}\boldsymbol{y}^\top \mathrm{d}q_0(\boldsymbol{y}) - \left(\int w_i \boldsymbol{y}\mathrm{d}q_0(\boldsymbol{y})\right)\left(\int w_i \boldsymbol{y}\mathrm{d}q_0(\boldsymbol{y})\right)^\top\right] \quad \text{(by Lemma 4)}
\end{aligned}
\tag{36}
$$

$\square$

This proof is also inherently the calculation of the Hessian of a log-convolution of a density, we provide the detailed derivation here for completeness.

### A.4. Proof of Proposition 4

*Proof.* Given fixed $(x_t, t)$, $\mathcal{L}$ is a quadratic form of $y_\theta$. To obtain the optimal $\bar{\boldsymbol{y}}_\theta$, we differentiate $\mathcal{L}$ and set this derivative equal to zero, resulting in the following

$$0 = \frac{\partial \mathcal{L}}{\partial \bar{\boldsymbol{y}}_\theta(\boldsymbol{x}_t, t)} = \frac{\partial}{\partial \bar{\boldsymbol{y}}_\theta(\boldsymbol{x}_t, t)} \int_{\mathbb{R}^d} \underbrace{\frac{1}{N}(2\pi\sigma_t^2)^{-\frac{d}{2}} v(\boldsymbol{x}_t, t, \boldsymbol{y}) \lambda_t}_{A_t} \frac{\alpha_t^2}{\sigma_t^2} \|\bar{\boldsymbol{y}}_\theta(\boldsymbol{x}_t, t) - \boldsymbol{y}\|^2 \mathrm{d}q_0(\boldsymbol{y})$$

$$= 2 A_t \lambda_t \frac{\alpha_t^2}{\sigma_t^2} \int_{\mathbb{R}^d} \boldsymbol{v}(x_t, t, \boldsymbol{y})(\bar{\boldsymbol{y}}_\theta(\boldsymbol{x}_t, t) - \boldsymbol{y}) \mathrm{d}q_0(\boldsymbol{y}), \tag{37}$$

which yields

$$\bar{\boldsymbol{y}}_\theta^*(\boldsymbol{x}_t, t) = \int_{\mathbb{R}^d} \frac{v(\boldsymbol{x}_t, t, \boldsymbol{y}')}{\int_{\mathbb{R}^d} v(\boldsymbol{x}_t, t, \boldsymbol{y}'') \mathrm{d}q_0(\boldsymbol{y}'')} \boldsymbol{y}' \mathrm{d}q_0(\boldsymbol{y}') = \int_{\mathbb{R}^d} w(\boldsymbol{x}_t, t, \boldsymbol{y}') \boldsymbol{y}' \mathrm{d}q_0(\boldsymbol{y}'). \tag{38}$$

$\square$

## A.5. Proof of Proposition 5

*Lemma* 5. Given a vector $\boldsymbol{v} \in \mathbb{R}^d$, the trace of the outer-product matrix of this vector is precisely equal to the square of its 2-norm. This can be shown as follows:

$$\mathrm{tr}\left(\boldsymbol{v}\boldsymbol{v}^T\right) = \sum_{i=1}^{d} \left(\boldsymbol{v}\boldsymbol{v}^T\right)_{i,i} = \sum_{i=1}^{d} v_i * v_i = \|\boldsymbol{v}\|^2 \tag{39}$$

*Proof.* Then we can start to give the derivation of Proposition 5

$$\mathrm{tr}\left(\boldsymbol{F}_t(\boldsymbol{x}_t, t)\right) = \mathrm{tr}\left(\frac{1}{\sigma_t^2}\boldsymbol{I} - \frac{\alpha_t^2}{\sigma_t^4}\left[\sum_i w_i \boldsymbol{y}_i \boldsymbol{y}_i^\top - \left(\sum_i w_i \boldsymbol{y}_i\right)\left(\sum_i w_i \boldsymbol{y}_i\right)^\top\right]\right)$$

$$= \frac{1}{\sigma_t^2}\mathrm{tr}\left(\boldsymbol{I}\right) - \frac{\alpha_t^2}{\sigma_t^4}\left[\sum_i w_i \mathrm{tr}\left(\boldsymbol{y}_i \boldsymbol{y}_i^\top\right) - \mathrm{tr}\left(\left(\sum_i w_i \boldsymbol{y}_i\right)\left(\sum_i w_i \boldsymbol{y}_i\right)^\top\right)\right] \tag{40}$$

$$= \frac{d}{\sigma_t^2} - \frac{\alpha_t^2}{\sigma_t^4}\left[\sum_i w_i \|\boldsymbol{y}_i\|^2 - \left\|\sum_i w_i \boldsymbol{y}_i\right\|^2\right]$$

$\square$

## A.6. Proof of Proposition 6

*Proof.* The objective in Algorithm 1 obviously equals to:

$$\arg\min_{t_\theta} \mathbb{E}_{\boldsymbol{x}_0 \sim q_0(\mathbf{x}_0), \boldsymbol{x}_t \sim \mathcal{N}(\alpha(t)\boldsymbol{x}_0, \sigma^2(t)\boldsymbol{I})} \left| t_\theta(\boldsymbol{x}_t, t) - \frac{\|\boldsymbol{x}_0\|^2}{d} \right|^2. \tag{41}$$

By expressing the expectation of Equation equation 41 in the form of a marginal distribution, we can transform the objective as follows:

$$\arg\min_{t_\theta} \sum_i \frac{1}{N}(2\pi\sigma_t^2)^{-\frac{d}{2}} \left| t_\theta(\boldsymbol{x}_t, t) - \frac{\|\boldsymbol{y}_i\|^2}{d} \right|^2 \tag{42}$$

The optimal $t_\theta^*$ must satisfy the condition that the gradient of the loss equals $0$. Therefore, we have:

$$
\begin{aligned}
0 &= \nabla_{t_\theta^*(\boldsymbol{x}_t,t)}\left[\sum_i \underbrace{\frac{1}{N}(2\pi\sigma_t^2)^{-\frac{d}{2}}}_{A_t} v_i(\boldsymbol{x}_t,t)\left|\frac{\|\boldsymbol{y}_i\|^2}{d}-t_\theta^*(\boldsymbol{x}_t,t)\right|^2\right] \\
&= \sum_i A_t v_i(\boldsymbol{x}_t,t)(t_\theta^*(\boldsymbol{x}_t,t)-\frac{\|\boldsymbol{y}_i\|^2}{d}) \\
&= A_t \sum_j v_j(\boldsymbol{x}_t,t)t_\theta^*(\boldsymbol{x}_t,t) - A_t\sum_i v_i(\boldsymbol{x}_t,t)\frac{\|\boldsymbol{y}_i\|^2}{d},
\end{aligned}
$$

Thus

$$
\begin{aligned}
t_\theta^*(\boldsymbol{x}_t,t) &= \frac{A_t \sum_i v_i(\boldsymbol{x}_t,t)\frac{\|\boldsymbol{y}_i\|^2}{d}}{A_t \sum_j v_j(\boldsymbol{x}_t,t)} \\
&= \sum_i \frac{v_i(\boldsymbol{x}_t,t)}{\sum_j v_j(\boldsymbol{x}_t,t)}\frac{\|\boldsymbol{y}_i\|^2}{d} \\
&= \frac{1}{d}\sum_i w_i(\boldsymbol{x}_t,t)\|\boldsymbol{y}_i\|^2
\end{aligned}
\tag{43}
$$

We have successfully completed the proof that the optimal $t_\theta(\boldsymbol{x}_t,t)$, as trained by Algorithm 1, is equivalent to $\frac{1}{d}\sum_i w_i(\boldsymbol{x}_t,t)\|\boldsymbol{y}_i\|^2$. $\qquad\square$

### A.7. Proof of Proposition 7

The approximation error of the estimated trace equation 17 will be its difference from the true Fisher information trace equation 16. We use consecutive Cauchy–Schwarz and triangle inequality to get the bound of the approximation error:

$$
\begin{aligned}
&\left|\frac{d}{\sigma_t^2}-\frac{\alpha_t^2}{\sigma_t^4}\left[\sum_i w_i\|\boldsymbol{y}_i\|^2-\left\|\sum_i w_i\boldsymbol{y}_i\right\|^2\right]-\left\{d\left[\frac{1}{\sigma_t^2}-\frac{\alpha_t^2}{\sigma_t^4}\left(t_\theta(\boldsymbol{x}_t,t)-\left\|\frac{\boldsymbol{x}_t-\sigma_t\boldsymbol{\varepsilon}_\theta(\boldsymbol{x}_t,t)}{\alpha_t}\right\|^2\right)\right]\right\}\right| \\
&=\frac{\alpha_t^2}{\sigma_t^4}\left|\sum_i w_i\|\boldsymbol{y}_i\|^2-\left\|\sum_i w_i\boldsymbol{y}_i\right\|^2-\left(t_\theta(\boldsymbol{x}_t,t)-\left\|\frac{\boldsymbol{x}_t-\sigma_t\boldsymbol{\varepsilon}_\theta(\boldsymbol{x}_t,t)}{\alpha_t}\right\|^2\right)\right| \\
&\leq\frac{\alpha_t^2}{\sigma_t^4}\left[\left|\sum_i w_i\|\boldsymbol{y}_i\|^2-t_\theta(\boldsymbol{x}_t,t)\right|+\left\|\sum_i w_i\boldsymbol{y}_i-\frac{\boldsymbol{x}_t-\sigma_t\boldsymbol{\varepsilon}_\theta(\boldsymbol{x}_t,t)}{\alpha_t}\right\|\right] \\
&\leq\frac{\alpha_t^2}{\sigma_t^4}\left[\delta_1+\frac{\sigma_t^2}{\alpha_t^2}\delta_2^2\right] \\
&=\frac{\alpha_t^2}{\sigma_t^4}\delta_1+\frac{1}{\sigma_t^2}\delta_2^2
\end{aligned}
\tag{44}
$$

## A.8. Proof of Proposition 8

$$
\begin{aligned}
&\left\| \frac{1}{\sigma_t} \boldsymbol{I} - \frac{\alpha_t^2}{\sigma_t^3} \left[ \sum_i w_i \boldsymbol{y}_i \boldsymbol{y}_i^\top - \left( \sum_i w_i \boldsymbol{y}_i \right) \left( \sum_i w_i \boldsymbol{y}_i \right)^\top \right] - \left( \frac{1}{\sigma_t} \boldsymbol{I} - \frac{\alpha_t^2}{\sigma_t^3} \left( \boldsymbol{x}_0 \boldsymbol{x}_0^\top - \bar{\boldsymbol{y}}_\theta(\boldsymbol{x}_t, t) \bar{\boldsymbol{y}}_\theta(\boldsymbol{x}_t, t)^\top \right) \right) \right\|_{HS} \\
&= \left\| -\frac{\alpha_t^2}{\sigma_t^3} \left[ \sum_i w_i \boldsymbol{y}_i \boldsymbol{y}_i^\top - \left( \sum_i w_i \boldsymbol{y}_i \right) \left( \sum_i w_i \boldsymbol{y}_i \right)^\top \right] - \left( -\frac{\alpha_t^2}{\sigma_t^3} \left( \boldsymbol{x}_0 \boldsymbol{x}_0^\top - \bar{\boldsymbol{y}}_\theta(\boldsymbol{x}_t, t) \bar{\boldsymbol{y}}_\theta(\boldsymbol{x}_t, t)^\top \right) \right) \right\|_{HS} \\
&\leq \frac{\alpha_t^2}{\sigma_t^3} \left\| \sum_i w_i \boldsymbol{y}_i \boldsymbol{y}_i^\top - \boldsymbol{x}_0 \boldsymbol{x}_0^\top \right\|_{HS} + \frac{\alpha_t^2}{\sigma_t^3} \left\| \left( \sum_i w_i \boldsymbol{y}_i \right) \left( \sum_i w_i \boldsymbol{y}_i \right)^\top - \bar{\boldsymbol{y}}_\theta(\boldsymbol{x}_t, t) \bar{\boldsymbol{y}}_\theta(\boldsymbol{x}_t, t)^\top \right\|_{HS} \\
&= \frac{\alpha_t^2}{\sigma_t^3} \sum_i w_i \left\| \boldsymbol{y}_i \boldsymbol{y}_i^\top - \boldsymbol{x}_0 \boldsymbol{x}_0^\top \right\|_{HS} + \frac{\alpha_t^2}{\sigma_t^3} \left\| \left( \sum_i w_i \boldsymbol{y}_i \right) \left( \sum_i w_i \boldsymbol{y}_i \right)^\top - \bar{\boldsymbol{y}}_\theta(\boldsymbol{x}_t, t) \bar{\boldsymbol{y}}_\theta(\boldsymbol{x}_t, t)^\top \right\|_{HS} \\
&\leq \frac{\alpha_t^2}{\sigma_t^3} \sum_i w_i \max_i \left\| \boldsymbol{y}_i \boldsymbol{y}_i^\top - \boldsymbol{x}_0 \boldsymbol{x}_0^\top \right\|_{HS} + \frac{\alpha_t^2}{\sigma_t^3} \sum_j \left| \sum_i w_i \boldsymbol{y}_i[j] - \bar{\boldsymbol{y}}_\theta(\boldsymbol{x}_t, t)[j] \right| \left\| \sum_i w_i \boldsymbol{y}_i - \bar{\boldsymbol{y}}_\theta(\boldsymbol{x}_t, t) \right\| \\
&\leq \frac{\alpha_t^2}{\sigma_t^3} \left( 2\mathcal{D}_y^2 + \sqrt{d}\delta_2 \right)
\end{aligned}
\tag{45}
$$

## A.9. Preliminaries on Optimal Transport

The optimal transport is the general problem of moving one distribution of mass to another as efficiently as possible. The *optimal transport problem* can be formulated in two primary ways, namely the Monge formulation (Monge, 1781) and the Kantorovich formulation (Kantorovich, 1960). Suppose there are two probability measures $\mu$ and $\nu$ on $(\mathbb{R}^n, \mathcal{B})$, and a cost function $c : \mathbb{R}^n \times \mathbb{R}^n \to [0, +\infty]$. The *Monge problem* is

$$
\text{(MP)} \qquad \inf_{\mathrm{T}} \left\{ \int c(x, \mathrm{T}(x)) \, \mathrm{d}\mu(x) : \mathrm{T}_\# \mu = \nu \right\}.
\tag{46}
$$

The measure $\mathrm{T}_\# \mu$ is defined through $\mathrm{T}_\# \mu(A) = \mu(\mathrm{T}^{-1}(A))$ for every $A \in \mathcal{B}$ and is called the *pushforward* of $\mu$ through $\mathrm{T}$.

It is evident that the Monge Problem (MP) transports the entire mass from a particular point, denoted as $x$, to a single point $\mathrm{T}(x)$. In contrast, Kantorovich provided a more general formulation, referred to as the *Kantorovich problem*:

$$
\text{(KP)} \qquad \inf_{\gamma} \left\{ \int_{\mathbb{R}^n \times \mathbb{R}^n} c \, \mathrm{d}\gamma : \gamma \in \Pi(\mu, \nu) \right\},
\tag{47}
$$

where $\Pi(\mu, \nu)$ is the set of *transport plans*, i.e.,

$$
\Pi(\mu, \nu) = \left\{ \gamma \in \mathcal{P}(\mathbb{R}^n \times \mathbb{R}^n) : (\pi_x)_\# \gamma = \mu, (\pi_y)_\# \gamma = \nu \right\},
\tag{48}
$$

where $\pi_x$ and $\pi_y$ are the two projections of $\mathbb{R}^n \times \mathbb{R}^n$ onto $\mathbb{R}^n$. For measures absolutely continuous with respect to the Lebesgue measure, these two problems are equivalent (Villani et al., 2009). However, when the measures are discrete, they are entirely distinct as the constraint of the Monge Problem may never be fulfilled.

## A.10. Proof of Corollary 1

To prove the Corollary 1, we first introduce two theorems to transform the problem of whether the PF-ODE mapping is a Monge map into the task of deciding the convexity of the potential function of $T_{s,T}$.

*Theorem* 1. (Santambrogio, 2015, Theorem 1.48) Suppose that $\mu$ is a probability measure on $(\mathbb{R}^n, \mathcal{B})$ such that $\int |x|^2 \mathrm{d}\mu(x) < \infty$ and that $u : \mathbb{R}^n \to \mathbb{R} \cup \{+\infty\}$ is convex and differentiable $\mu$-a.e. Set $\mathrm{T} = \nabla u$ and suppose $\int |\mathrm{T}(x)|^2 \mathrm{d}\mu(x) < \infty$. Then $\mathrm{T}$ is optimal for the transport cost $c(x, y) = \frac{1}{2}|x - y|^2$ between the measures $\mu$ and $\nu = \mathrm{T}_\# \mu$.

> *Theorem* 2. **The Brenier's Theorem.** (Santambrogio, 2015, Theorem 1.22) (Brenier, 1987; 1991) Let $\mu, v$ be probabilities over $\mathbb{R}^d$ and $c(x, y) = \frac{1}{2}|x - y|^2$. Suppose $\int |x|^2 \, dx, \int |y|^2 \, dy < +\infty$, which implies $\min(\text{KP}) < +\infty$ and suppose that $\mu$ gives no mass to $(d-1)$ surfaces of class $C^2$. Then there exists, uniquely, an optimal transport map T from $\mu$ to $v$, and it is of the form $T = \nabla u$ for a convex function $u$.

To ensure the existence of the potential function, we need to leverage the following

> *Theorem* 3. **The Poincaré's Theorem.** (Lang, 2012, Theorem 4.1 of Chapter V, §4) Let $U$ be an open ball in $\mathbb{R}^n$ and let $\omega$ be a differential form of degree $\geq 1$ on $U$ such that $d\omega = 0$. Then there exists a differential form $\phi$ on $U$ such that $d\phi = \omega$.

> *Remark* 1. The conclusion remains valid when the open ball $U$ is substituted with the entirety of $\mathbb{R}^n$

For $s > 0$, it is clear that $\frac{dq_T(x_T)}{dq_s(x_s)}$ is non-singular, and therefore, it satisfies the requirements of Brenier's Theorem 2, leading to the existence of a unique optimal transport map. According to Theorem 1, if we can establish that the potential function of $T_{s,T}$ is convex, then the PF-ODE mapping will indeed be a Monge map. Notice that the existence of the potential map is guaranteed by Poincaré's Theorem 3.

We can now convert the condition of the potential function of $T_{s,T}$ being convex into the condition that its Jacobian, $\frac{\partial T_{s,T}(x_s)}{\partial x_s}$, is positive semi-definite, as per the following theorem in convex analysis.

> *Theorem* 4. (Rockafellar, 2015, Theorem 4.5) Let $f$ be a twice continuously differentiable real-valued function on an open convex set $C$ in $R^n$. Then $f$ is convex on $C$ if and only if its Hessian matrix
>
> $$Q_x = (q_{ij}(x)), \quad q_{ij}(x) = \frac{\partial^2 f}{\partial \xi_i \partial \xi_j}(\xi_1, \dots, \xi_n) \tag{49}$$
>
> is positive semi-definite for every $x \in C$.

If we denote that

$$\boldsymbol{A}(t) = \frac{\partial T_{t,T}(x_t)}{\partial x_t} \tag{50}$$

obviously, $\boldsymbol{A}(T) = I$ is p.s.d., our goal is to answer when $\boldsymbol{A}(t)$ is p.s.d.. We try to answer this to set up a connection between $\boldsymbol{A}(t)$ and $\boldsymbol{A}(T) = I$. We can derive that:

$$
\begin{aligned}
\frac{d\boldsymbol{A}(t)}{dt} &= \lim_{\epsilon \to 0^+} \frac{\boldsymbol{A}(t+\epsilon) - \boldsymbol{A}(t)}{\epsilon} \\
&= \lim_{\epsilon \to 0^+} \frac{\boldsymbol{A}(t+\epsilon) - \boldsymbol{A}(t+\epsilon)\nabla_{\boldsymbol{x}_t}\left[\boldsymbol{x}_t + \epsilon\left(f(t)\boldsymbol{x}_t - \frac{g^2(t)}{2}\nabla_{\boldsymbol{x}_t}\log_t q_t(\boldsymbol{x}_t)\right) + \mathcal{O}(\epsilon^2)\right]}{\epsilon} \\
&= \lim_{\epsilon \to 0^+} \frac{\epsilon f(t)\boldsymbol{A}(t+\epsilon) + \frac{g^2(t)}{2}\epsilon\boldsymbol{A}(t+\epsilon)\nabla_{\boldsymbol{x}_t}\log_t q_t(\boldsymbol{x}_t) + \mathcal{O}(\epsilon^2)}{\epsilon} \\
&= f(t)\boldsymbol{A}(t) + \frac{g^2(t)}{2}\boldsymbol{A}(t)\nabla_{\boldsymbol{x}_t}\log_t q_t(\boldsymbol{x}_t) \\
&= f(t)\boldsymbol{A}(t) - \frac{g^2(t)}{2}\boldsymbol{A}(t)\left\{\frac{1}{\sigma_t^2}\boldsymbol{I} - \frac{\alpha_t^2}{\sigma_t^4}\left[\int w(\boldsymbol{y})\boldsymbol{y}\boldsymbol{y}^\top dq_0(\boldsymbol{y}) - \left(\int w(\boldsymbol{y})\boldsymbol{y}dq_0(\boldsymbol{y})\right)\left(\int w(\boldsymbol{y})\boldsymbol{y}dq_0(\boldsymbol{y})\right)^\top\right]\right\} \\
&= \boldsymbol{A}(t)\underbrace{\left\{\left[f(t) - \frac{g^2(t)}{2\sigma_t^2}\right]\boldsymbol{I} + \frac{\alpha_t^2 g^2(t)}{2\sigma_t^4}\left[\sum_i w_i\boldsymbol{y}_i\boldsymbol{y}_i^\top - \left(\sum_i w_i\boldsymbol{y}_i\right)\left(\sum_i w_i\boldsymbol{y}_i\right)^\top\right]\right\}}_{\boldsymbol{B}(t)}
\end{aligned}
$$

$$\tag{51}$$

Notice that the above ODE starts from $T$.

According to the Solution Matrices theory (Masuyama, 2016)[1], let us denote the $C(t)$ is the normalized fundamental matrix at $T$ for $\boldsymbol{B}(T)$, which implies $C(t)$ is the solution to the following ODE:

$$\boldsymbol{C}'(t) = \boldsymbol{C}(t)\boldsymbol{B}(t), \boldsymbol{C}(T) = I, \quad \text{(flow from T to t)} \tag{52}$$

Then we can deduce that

$$\begin{aligned}
\frac{\partial T_{t,T}(x_t)}{\partial x_t} &= \boldsymbol{A}(t) \\
&= \boldsymbol{C}(t)\boldsymbol{A}(T) \\
&= \boldsymbol{C}(t)\boldsymbol{I} \\
&= \boldsymbol{C}(t)
\end{aligned} \tag{53}$$

Thus, the diffeomorphism $T_{s,T}$ is a Monge optimal transport map if and only if $\boldsymbol{C}(s)$ is semi-positive definite. Note that the above requirement needs to be satisfied for every PF-ODE chain $x_t, t \in [T, t]$.

# B. Experiments Details

### B.1. Evaluation of NLL

For toy experiments, we follow the experiment design in Lu et al. (2022a) and adopt the same network architecture for the score network and our DF-TM network. We calculate the true diffusion Fisher with 5,000 data samples from the cheeseboard distribution.

For commercial-level experiments, we employ the explicit Euler method to compute the NLL, excluding the final step near $t = 0$, for reasons discussed in Appendix C.1. We evaluate the NLL across 10 steps throughout the timeline of the PF-ODE. The DF-TM network we trained uses the float-point16 data type.

**Network architectures** In terms of network architecture, we employ an SD Unet structure with an additional MLP head. However, we believe that a lighter network could potentially be sufficient for DF-TM.

**Training cost of DF-TM** For training the DF-TM network, we approximately spend 24 hours using 8 Tesla V100 chips. Given the size of the Laion2B-en dataset (which contains 2.32 billion images), this is quite an efficient speed. Additionally, the convergence behavior of the training loss is robust, as illustrated in Figure 1a. We also hypothesize that our network design has redundancy, suggesting that we could further reduce costs by opting for lighter networks. We will provide more details about training costs in our revised manuscript.

### B.2. Adjoint Guidance Sampling

For Figure 3, we examined varying numbers of adjoint guidance, ranging from 0 to 20, under a full inference number of 50. The adjoint guidance scale was grid-searched by the VJP method. For experiments on Pick-Score, we use NVIDIA V100 chips, and the rest experiments use Tesla V100 chips.

**Base method for DF-EA** Several variants of adjoint-optimization algorithms exist, such as AdjointDPM (Pan et al., 2023a), AdjointDES (Blasingame & Liu, 2024), and SAG (Pan et al., 2023b). However, while these algorithms differ in their design of solvers for the adjoint ODE, they all utilize VJP when accessing the diffusion Fisher. We selected SAG as our base method due to its state-of-the-art performance. We believe that replacing VJP with DF-EA could also enhance the performance of algorithms like AdjointDPM and AdjointDES.

**The Design of Score functions**

- Aesthetic Score (SAC/AVA/Pick-Score)

---

[1]The Definition 6.2 in `https://math.mit.edu/~jorloff/suppnotes/suppnotes03/ls6.pdf` suffice the result here.

For the aesthetic score predictor $f_{aes} : \mathbb{R}^d \mapsto \mathbb{R}$, the adjoint optimization target is simply $f_{aes}$ itself.

$$\mathcal{L}(\boldsymbol{x}_0) := f_{aes}(\boldsymbol{x}_0) \tag{54}$$

- Clip Loss

  Following the implementation of (Pan et al., 2023a), we use the features from the CLIP image encoder as our feature vector. The loss function is $L_2$-norm between the Gram matrix of the style image and the Gram matrix of the estimated clean image.

$$\mathcal{L}(\boldsymbol{x}_0) := \left\| \mathrm{clip}(\boldsymbol{x}_0)\mathrm{clip}(\boldsymbol{x}_0)^\top - \mathrm{clip}(\boldsymbol{x}_{ref})\mathrm{clip}(\boldsymbol{x}_{ref})^\top \right\| \tag{55}$$

- FaceID Loss

  Following the implementation of (Pan et al., 2023b), we use ArcFace to extract the target features of reference faces to represent face IDs and compute the $l_2$ Euclidean distance between the extracted ID features of the estimated clean image and the reference face image as the loss function.

$$\mathcal{L}(\boldsymbol{x}_0) := \left\| \mathrm{ArcFace}(\boldsymbol{x}_0) - \mathrm{ArcFace}(\boldsymbol{x}_{ref}) \right\| \tag{56}$$

**Hyperparameters** For the hyperparameters in adjoint-guided sampling, we ensure a fair comparison between the VJP and DF-EA methods. For most hyperparameters, we directly adopt the settings from previous works (Pan et al., 2023b) for both the baseline VJP method and our DF-EA method. For the guidance scale, we tune the value for the VJP method and use the same value for our DF-EA method. The DF-EA method does not introduce additional hyperparameters, and for mutual hyperparameters, our DF-EA method uses the exact same values as the VJP method. We adopt this strategy because our DF-EA method solely improves the approximation of the diffusion Fisher linear operator, without altering the adjoint sampling mechanism. Therefore, the suitable hyperparameters should remain unchanged, and we simply use the same parameters from the VJP method for our DF-EA method. For all experiments, we set the number of sampling steps to $T = 50$. Adjoint guidance is applied starting from steps ranging from 15 to 35 and ending at step 35, with one guidance per step. Thus the only parameter we tune is the guidance strength. We determine this value for the VJP method via a grid search from 0.1 to 0.5 with a step size of 0.1 and find that the optimal guidance strength for VJP is 0.2. We then use this value for our DF-EA method. Notice that, we apply a normalization to the guidance gradient for both VJP and DF-EA methods, making our optimal guidance strength consistent across different scores. The tuning is conducted on 1k COCO prompts, and the computational budget for tuning is 4 * 5 * 3 GPU hours (4 tasks * 5 grids * 3 hours per single test) on Tesla V100 chips.

### B.3. Numerical OT Experiments

**The detailed algorithm** The detailed version of the Algorithm 2 is presented in Algorithm 3. We use all the data points in the initial dataset to calculate every quantum we need in the sampling trajectory to validate the condition discussed in Corollary 1.

**Initial data** In Table 3, we adopt numerical verification of OT on 2-D synthesized data. For the single-Gaussian case, we fix the one mode central in (0.5,0.5) and test at $s = 0.1$; For the affine data case, we set three data points as (0.2,-0.4), (0.2,0.0), and (0.2,0.9), and test at $s = 0.0$; For the non-affine data case, we set three data points as (0.0,0.5), (0.0,0.0), and (0.5,0.0), and test at $s = 0.0$.

**The definition of Asymmetric rate** In Table 3, we need the asymmetry rate $I$ to point out that the PF-ODE map in very simple non-affine data cannot be OT. Let $A$ be an $n \times n$ matrix. The asymmetry rate $I$ is defined as:

$$I = \frac{\left\| A - A^T \right\|_F}{\sqrt{2}\|A\|_F}$$

where $\|M\|_F = \sqrt{\sum_{i=1}^n \sum_{j=1}^n m_{ij}^2}$ is the Frobenius norm of the matrix $M$. The denominator $\sqrt{2}\|A\|_F$ is used to normalize the index to the range of $[0, 1]$. The asymmetry rate of the non-affine data in Table 3 has significantly deviated from zero.

---

**Algorithm 3** Detailed numerical OT test for PF-ODE map

---

1: **Input**: initial data $\{\boldsymbol{y}_j\}_{j=1}^N$, noise schedule $\{\alpha(t)\}$ and $\{\sigma(t)\}$, discretization steps $M$.
2: Initialize $\boldsymbol{A}_M = \boldsymbol{I}$, $\boldsymbol{x}_M \sim \mathcal{N}(0, \sigma_T \boldsymbol{I})$.
3: **for** $i = M, M-1, \cdots, 1$ **do**
4:     $\alpha_i = \alpha(t_i), \quad \sigma_i = \sigma(t_i)$
5:     $\mathrm{d}t = t_{i-1} - t_i$.
6:     $f_i = \frac{\mathrm{d}\log\alpha_i}{\mathrm{d}t}$.
7:     $g_i = \sqrt{\frac{\mathrm{d}\sigma_i^2}{\mathrm{d}t} - 2\frac{\mathrm{d}\log\alpha_i}{\mathrm{d}t}\sigma_i^2}$.
8:     **for** $j = 1, \cdots, N$ **do**
9:         calculate $v_j$ via $v_j = \exp\left(-\frac{|\boldsymbol{x}_i - \alpha_i \boldsymbol{y}_j|^2}{2\sigma_i^2}\right)$
10:    **end for**
11:    **for** $j = 1, \cdots, N$ **do**
12:        calculate $w_j$ via $w_j = \frac{v_j}{\sum_k v_k}$.
13:    **end for**
14:    $\boldsymbol{B}_i = \left[f_i - \frac{g_i^2}{2\sigma_i^2}\right]\boldsymbol{I} + \frac{\alpha_i^2 g_i^2}{2\sigma_i^4}\left[\sum_j w_j \boldsymbol{y}_j \boldsymbol{y}_j^\top - \left(\sum_j w_j \boldsymbol{y}_j\right)\left(\sum_j w_j \boldsymbol{y}_j\right)^\top\right]$.
15:    $\boldsymbol{A}_{i-1} = \boldsymbol{A}_i + \mathrm{d}t * \boldsymbol{A}_i^\top \boldsymbol{B}_i$                    {solve fundamental matrix}
16:    $s_i = \frac{\boldsymbol{x}_i - \sum_j w_j y_j}{\sigma_i^2}$
17:    $\boldsymbol{x}_{i-1} = \boldsymbol{x}_i + \left[f_i \boldsymbol{x}_i - \frac{1}{2}g_i^2 s_i\right] * \mathrm{d}t$
18: **end for**
19: **Output**: $\boldsymbol{A}_0$.

---

## B.4. Pretrained Models

All of the pretrained models used in our research are open-sourced and available online as follows:

- stable-diffusion-v1-5

  `https://huggingface.co/runwayml/stable-diffusion-v1-5`

- stable-diffusion-2-base

  `https://huggingface.co/stabilityai/stable-diffusion-2-base`

- SAC-aesthetic score predictor

  `https://github.com/christophschuhmann/improved-aesthetic-predictor/blob/main/sac%2Blogos%2Bava1-l14-linearMSE.pth`

- AVA-aesthetic score predictor

  `https://github.com/christophschuhmann/improved-aesthetic-predictor/blob/main/ava%2Blogos-l14-linearMSE.pth`

- ArcFace ID loss

  `https://github.com/TreB1eN/InsightFace_Pytorch`

- Clip loss

  `https://huggingface.co/openai/clip-vit-large-patch14`

- Pick-Score

  `https://github.com/yuvalkirstain/PickScore`

# C. Discussions

### C.1. Singularity of Fisher information at $t = 0$

Previous studies (Yang et al., 2023; Zhang et al., 2024b) have shown that the diffusion model, particularly when learned in $\epsilon$-prediction, can encounter a singularity issue at $t = 0$. Our DF in equation 11 reaffirms this issue, as this formulation becomes ill-formed at $t = 0$ due to division by zero ($\sigma_0$). Consequently, our formulation does not describe the behavior at $t = 0$. The deep theoretical exploration of the singularity problem remains an open question in the diffusion model field. However, as it is not the primary focus of this paper, we will not discuss the DF at $t = 0$.

### C.2. Statistical Caliber of Negative Log-Likelihood

When dealing with high-dimensional data such as images, direct likelihood comparisons may encounter scaling issues due to the dimensionality. In this study, unless explicitly indicated otherwise, we adopt the approach of (Zheng et al., 2023) and typically use Negative Log-Likelihood (NLL) to refer to Bits Per Dimension (BPD).

$$\text{BPD} = \mathbb{E}_{\boldsymbol{x}_0 \sim q_0} \left[ \frac{-\log P_0\left(\boldsymbol{x}_0\right)}{d \log 2} \right] \tag{57}$$

### C.3. The ODE Solvers

To compute the numerical solutions for the PF-ODE in equation 5, the likelihood ODE in equation 14, and the adjoint ODE in equation 18, we require ODE solvers. In our paper's experiments, we consistently use the explicit Euler method (referred to as DDIM when applied to PF-ODE). However, it's important to note that our approach is not dependent on a specific ODE solver. We can also utilize alternatives like fast ODE solvers (Lu et al., 2022b;c; Liu et al., 2022) or exact inversion ODE solvers (Wallace et al., 2023; Zhang et al., 2023).

### C.4. The relation of our outer-product form DF to Covariance in DMs

We note that there is a series of studies aiming to learn the covariance of the reverse diffusion Stochastic Differential Equation (SDE) (Bao et al., 2022b;a). In Bayesian statistics, Fisher information is defined as the covariance of the score. These studies derive their formulation by analyzing the covariance of the score and obtaining the Fisher information in terms of the score. However, our DF is derived directly from the marginal distribution and is composed solely of the initial distribution and noise schedule. Furthermore, our application is unique; we are the first to replace the use of VJP with DF, while the focus of these studies is to enhance the performance of Diffusion Models (DMs) with analytical covariance.

A very similar form of the Fisher information is proposed in Lemma 5 of (Benton et al., 2024). The difference is that our Proposition 3 presents the specific form of Fisher information in terms of data distribution, which is not included in Lemma 5 of (Benton et al., 2024). This distinction is crucial as it facilitates the derivation of our new algorithms. Also, we adopt the currently more commonly used $\alpha_t, \sigma_t$ notation to represent the noise schedule. Instead, (Benton et al., 2024) $\alpha_t \equiv e^{-2t}$. We gave out this detailed formulation to avoid unnecessary misunderstandings in the development of the subsequent training scheme.

### C.5. Comparison of DF-TM to Hutchinson trace estimator

We notice that the naive trace calculation of the VJP method can be accelerated by the Hutchinson trace estimator. We have not conducted experiments using the Hutchinson estimator, as it is a Monte-Carlo type estimation, unlike the direct evaluation methods like the full-VJP baseline and our DF-TM. Furthermore, the VJP method, with the help of the Hutchinson method and its variants, is still considerably more expensive than our method, and its applicability is also more restricted.

- **The Hutchinson method is much more costly:**

  To attain a relative error less than $\epsilon$ with a probability of $1 - \delta$, the Hutchinson method requires $\frac{2(1-\frac{8}{3}\epsilon)\log\frac{1}{\delta}}{\epsilon^2}$ samples (Skorski, 2021). Assuming that the goal is to obtain an estimation with a relative error of less than 10% with a probability exceeding 90%, a minimum of 351 NFEs is required. This equates to 1 or 2 minutes on SD-v1.5 for a single trace estimation. For a complete NLL estimation of an image with 20 steps, this would take 20 minutes, which is entirely impractical in any business context. In contrast, our DF-TM only requires 1 NFE for a single trace estimation, needing merely 10 seconds for the full NLL estimation of an image.

- **The application of the Hutchinson method is more restricted:**

  Due to its Monte-Carlo characteristics, the Hutchinson method is more appropriate for contributing to the computation of certain training objectives, as in (Lu et al., 2022a), where the unbiased property is sufficient, and large variance may be absorbed into network training. However, the Hutchinson method may encounter difficulties with accurate per-sample trace estimation. Our method can accommodate both scenarios, including per-sample computation.

There are already attempts to use the Hutchinson method to expedite the VJP of trace estimation in diffusion models (Lu et al., 2022a). Nonetheless, due to Hutchinson's limitations, these practices are restricted to relatively small DMs (CIFAR-10 at most). We will add discussions on the Hutchinson method in our revision.

### C.6. Discussions on the theoretical bound of DF-EA

We notice that the error bound in Proposition 8 does not vanish as the training error decreases. From the rigorous theoretical perspective, considering the error bounds in Propositions 7 and 8, the DF-EA method is less valid than the DF-TM method, as we currently cannot establish vanishing bounds for it. However, from an empirical perspective, our DF-EA outperforms the naive VJP method in terms of score improvement across various tasks and pretrained models, as demonstrated in Figure 4. Thus, the DF-EA approximation proves to be valid in a practical sense. The replacement $\sum_i \boldsymbol{y}_i \boldsymbol{y}_i^\top \approx \boldsymbol{x}_0 \boldsymbol{x}_0^\top$ originates from the observation that $w_i$ is a weighting of the summation equal to 1, and as $t$ approaches 0, the $w_i$ closest to $\boldsymbol{x}_0$ will dominate due to the diffusion kernel. Therefore, this approximation is intuitively reasonable near $t = 0$, which is precisely where we apply adjoint guidance.

### C.7. Limitations

This paper does not explore the integration of DF into accelerated ODE solvers (Wang et al., 2025; Lu et al., 2022b), or exact inversion ODE samplers (Wang et al., 2024a). This paper is constrained in the scope of DMs, but similar second-order information may also exist in flow-matching generative models (Lipman et al., 2022; Zhu et al., 2024a) or variational inference models (Zhu et al., 2024b; Wang et al., 2024b). Our technique may also have downstream applications in automated driving (Tu et al., 2025b), touch-generation (Tu et al., 2025a), domain-transfer (Feng et al., 2024b;a), language generation (Fu et al., 2025), and missing data imputation (Chen et al., 2024). The DF may also contribute to a more effective inference method, like L-BFGS, which we did not explore. The complex second-order information in auto-regressive models can also be analytically explored (Cheng et al., 2025).

