# OpenReview forum: "Efficiently Access Diffusion Fisher: Within the Outer Product Span Space"
_ICML.cc/2025/Conference — ICML 2025 poster_

### Official Review · Reviewer_a1EQ · 2025-03-13

**Overall Recommendation:** 5

**Summary:**

The diffusion Fisher information matrix (or just 'diffusion Fisher') provides useful quantitative information about the sensitivity of diffusion model log-likelihoods to small changes in state, and can be exploited to (among other things) evaluate sample likelihoods and guide diffusion models to generate higher-quality samples.

The authors observe that a special property of the diffusion Fisher allows it to be computed more efficiently than is possible with the naive auto-differentiation-based approach. Moreover, they show that their approach is much faster than an existing alternative approach for computing (the trace of) the diffusion Fisher, which involves taking the Jacobian-vector product (JVP).

**Claims And Evidence:**

Yes. The authors' work is based on a simple theoretical idea, which they clearly prove. They support their claim of practical usefulness with a variety of numerical experiments, which involve both evaluating likelihoods and guiding the output of diffusion models to be higher-quality.

**Essential References Not Discussed:**

No references come to mind.

**Experimental Designs Or Analyses:**

The authors' experiments appear sound and yield reasonable-looking results.

**Methods And Evaluation Criteria:**

Yes, the authors' methods and evaluations (principally, using the diffusion Fisher for what it's usually used for, and showing that their computations are fast compared to the previous SOTA) make sense.

**Other Comments Or Suggestions:**

line ~ 208, "detail discussions" -> "a detailed discussion"

**Other Strengths And Weaknesses:**

The paper is well-written and the main theoretical idea is clear. The authors' evaluations are easy to understand and clear, although it would be helpful if more examples (along the lines of Figs 2 and 4) were shown in the SI.

**Questions For Authors:**

The authors' proposed methods involve some approximation (see, e.g., line 375). Are there cases where these approximations are expected to break down, and this method is not expected to give good results? In general, are there any obvious issues associated with using or scaling up these methods for computing the diffusion Fisher (or certain related quantities, like its trace)? As a related point, Proposition 6 appears to show potentially bad behavior when $\sigma_t \approx 0$.

**Relation To Broader Scientific Literature:**

The authors' work relates to the diffusion Fisher, which has various well-known applications to evaluating likelihoods and guiding diffusion models. More broadly, it relates to a large body of work concerning evaluating the quality of the samples of diffusion models, and guiding them to produce better samples.

**Theoretical Claims:**

The authors' theoretical claims are simple and easy to verify.

---

> ### Author Rebuttal · Authors · 2025-03-26
>
> Thank you for your appreciation of our work. We will answer your questions one by one regarding these suggestions/questions.
>
> > **Answer to Suggestion 1:**  The authors' evaluations are easy to understand and clear, although it would be helpful if more examples (along the lines of Figs 2 and 4) were shown in the SI.
>
> While we consider the theoretical analysis of the diffusion Fisher to be the primary contribution of this work, we agree that additional visual illustrations, such as those in Figures 2 and 4, would further enhance clarity. We will include more examples in the Supplementary Information (SI) as part of our revision.
>
>
> > **Answer to Suggestion 2:** line ~ 208, "detail discussions" -> "a detailed discussion"
>
> We apologize for this oversight. We have now corrected the syntax error and conducted a thorough proofreading of our work.
>
>
> > **Answer to Question 1:** The authors' proposed methods involve some approximation (see, e.g., line 375). Are there cases where these approximations are expected to break down, and this method is not expected to give good results?
>
> In the DF-EA method, we introduce an approximation technique for the adjoint term (around line 375). These approximations are likely to be inaccurate in the following scenarios:
> - When $t$ is very large, the $x$-prediction part in the approximation is less accurate because of the large noise scale.
> - When $t$ is very small, the approximation error bound stated in Proposition 6 will increase due to the singularity issue, and the approximation will lack an accuracy guarantee. We will discuss on this singularity issue in detail in Question 3
>
> In practice, we apply DF-EA in the range from $t=0.7$ to $t=0.3$. This range also aligns with the recommended adjoint region of the original adjoint methods.
>
> > **Answer to Question 2:** In general, are there any obvious issues associated with using or scaling up these methods for computing the diffusion Fisher (or certain related quantities, like its trace)?
>
> - Obvious issue: Our methods are derived exclusively on the classical diffusion process and thus cannot be directly extended to settings where the diffusion process is modified, such as cases involving additional consistency-model-type distillation [1], Schrödinger-Bridge-type endpoint alignment [2], non-Gaussian transition kernels [3], or flow rectification [4].
>
> - Scaling-up:
> Theoretically, both the training and sampling phases of our ​DF-TM and ​DF-EA exhibit ​linear complexity growth with respect to dataset scale and data dimensionality.
> Experimentally, our approach performs well on commercial-grade diffusion models such as SD2-base and large-scale datasets like Laion-2B. Therefore, our method indeed has scalability.
>
> [1] Frans, Kevin, et al. "One step diffusion via shortcut models." ICLR 2025.
>
> [2] De Bortoli, Valentin, et al. "Diffusion schrödinger bridge with applications to score-based generative modeling." NeurIPS 2022.
>
> [3] Yoon, Eun Bi, et al. "Score-based generative models with Lévy processes." NeurIPS 2023.
>
> [4] Liu, Xingchao, et al. Flow Straight and Fast: Learning to Generate and Transfer Data with Rectified Flow. ICLR 2022.
>
>
> > **Answer to Question 3:** As a related point, Proposition 6 appears to show potentially bad behavior when $\sigma_t \approx 0$.
>
> Note that Proposition 6 will be ill-defined due to division-by-zero when $t=0$ since $\sigma_0 = 0$. This concern is also raised by Reviewer V5LL. We argue that this problem, which has been referred to as the **singularity issue** in previous literature [1][2], stems from the inherent characteristics of diffusion models rather than flaws in our derivations.
>
> Theoretically, the diffusion model has an inherent singularity property when $t_{\min} \to 0$. This implies that many quantities associated with diffusion models become ill-defined in the vicinity of $t_{\min} = 0$. This phenomenon, previously explored in [1] and [2], is once again recognized in the context of diffusion Fisher through our analysis. A deep theoretical exploration of the singularity problem remains an open question within the diffusion model field.
>
> Practically, we implement the approach used in [3] and [4], where the diffusion sampling path is ended at $t_{\min} = 0.005$ instead of $t_{\min} = 0$. This method circumvents potential bad effects resulting from the singularity issue.
>
> We briefly discussed this singularity issue on line 1242 of the original paper. In our revision, we will provide a more detailed discussion and clarification of our handling of this issue.
>
> [1] Zhang, Pengze, et al. "Tackling the singularities at the endpoints of time intervals in diffusion models." CVPR 2024.
>
> [2] Yang, Zhantao, et al. "Lipschitz singularities in diffusion models." ICLR 2024.
>
> [3] Song Y, Dhariwal P, Chen M, et al. Consistency Models ICML 2023.
>
> [4] Lu, Cheng, et al. "Maximum likelihood training for score-based diffusion odes by high order denoising score matching." ICML 2022.

---

### Official Review · Reviewer_mnN8 · 2025-03-14

**Overall Recommendation:** 3

**Summary:**

This paper introduces a novel formulation of the diffusion Fisher (DF) information in diffusion models by expressing it as a weighted sum of outer products of the score function and initial data, thereby revealing that DF lies in a space spanned by specific outer-product bases dependent solely on the initial distribution and noise schedule. This formulation enables the development of two efficient approximation algorithms—one for computing the trace of DF and another for its matrix-vector multiplication—with rigorously established error bounds, significantly reducing computational costs compared to traditional auto-differentiation methods. Additionally, the paper derives a corollary for the optimal transport property of the diffusion-ODE deduced map and validates this property through numerical experiments on various noise schedules, demonstrating enhanced accuracy in tasks like likelihood evaluation and adjoint optimization.

**Claims And Evidence:**

I have a problem with the paper's main claim that JVP cannot be calculated effectively, which motivates the authors to train an additional network. More specifically, they claim that the time complexity is $O(d^2)$ and infeasible. However, both Pytorch and JAX provide forward-mode autodiff, which can be used to calculate JVP very efficiently (e.g. Pytorch with torch.func). Loosely, it requires only 2 forward passes, which is faster than a single optimization step (1 forward, 1 backward at least) on the same network. If the model evaluation is required anyway, then the overhead is just 1 forward pass. Thus, training a network to calculate the JVP seems unfavorable and unscalable. Even without forward-mode autodiff, how about approximating it with numerical methods? They may not be accurate, but I think it is a reasonable baseline that needs presenting.

**Essential References Not Discussed:**

Not that I know of.

**Experimental Designs Or Analyses:**

I do not have problems with them.

**Methods And Evaluation Criteria:**

They make sense to me.

**Other Comments Or Suggestions:**

Given my concerns stated before, I cannot recommend acceptance of this paper. However, there are claims that I enjoy learning, e.g. Prop 1, and the paper is well-written and easy to follow.

**Other Strengths And Weaknesses:**

Even without considering the problems listed in Claims And Evidence. Training a separate nontrivial network for the use cases listed in the paper feels too heavy. I suggest the authors justify the additional cost better with more practical and popular use cases.

**Questions For Authors:**

1. The adjoint optimization is an interesting topic, which I do not know very much about. Is it true that the prior methods all use the highly inefficient way of calculating the JVP? I wish the authors could go into more detail here, as I think this setting/use case has the most potential in practice.

**Relation To Broader Scientific Literature:**

Authors proposed a new way to calculate the trace of diffusion Fisher.

**Theoretical Claims:**

I checked the theoretical claims, though not their proofs in detail. I do not find any problems.

---

> ### Author Rebuttal · Authors · 2025-03-28
>
> Thank you for your valuable feedback! We will answer your concerns/questions one by one.
>
> > **Concern 1:** I have a problem with the paper's main claim that JVP cannot be calculated effectively, they claim that the time complexity is $O(d^2)$...
>
>
> Please allow me to humbly clarify that we didn't claim a single JVP's time complexity to be $O(d^2)$. Instead, we stated that calculating the diffusion Fisher trace using JVPs has an $O(d^2)$ complexity.
> The reason is that to obtain each diagonal element of the diffusion Fisher matrix, one JVP operation is needed. As computing the full trace demands all $d$ diagonal elements and each JVP has $O(d)$ complexity, the total complexity for $d$ JVPs is $O(d^2)$.
>
> In the context of accessing the diffusion Fisher trace, previous methods rely on $d$ JVP operations. In contrast, our DF-TM method only requires one forward pass through the learned trace network to access the diffusion Fisher trace, which is notably more efficient than the traditional multiple-JVP approach.
>
> In the adjoint scenario, previous methods use a JVP to compute the adjoint term. Our DF-EA approximation method eliminates this JVP, making it marginally more efficient than the traditional method.
>
> Furthermore, accessing the diffusion Fisher via JVP is a black-box approach without any accuracy guarantee. Conversely, our DF-TM and DF-EA methods exploit the analytical structure of the diffusion Fisher and offer theoretical guarantees.
>
>
> > **Concern 2:**  how about approximating it with numerical methods?...
>
> While numerical methods for JVP and trace evaluation are inaccurate and inefficient in high-dimensional scenarios, as discussed in line ~1281.
> We appreciate your suggestion and agree that incorporating these as baselines enhances the clarity of our experiments.
> So, we've added two numerical methods, Finite Difference and Hutchinson's Trace Estimation, as baselines in the toy NLL evaluation experiment:
>
> **Table 2: The relative error of NLL evaluation.**
> |Methods|t = 1.0|t = 0.8|t = 0.6|t = 0.4|t = 0.2|t = $t_{\min}$|
> |-|-|-|-|-|-|-|
> |Finite Difference|20.54%|44.57%|60.68%|79.53%|83.08%|94.87%|
> |Hutchinson|11.28%|10.01%|16.12%|23.79%|53.02%|71.85%|
> |JVP|6.68%|5.79%|10.46%|20.13%|51.14%|70.95%|
> |DF-TM ​(**Ours**​)|​**3.41%​**|​**4.56%​**|**4.13%​**|​**4.28%​**|​**5.33%​**|**5.81%​**|
>
> It is shown that our DF-TM outperforms these numerical methods.
>
> > **Concern 3:** Training a separate nontrivial network for the use cases listed in the paper feels too heavy...
>
> - In adjoint applications, our DF-EA is training-free.
> - In NLL evaluation, our DF-TM incorporates an additional trace matching network, only bringing moderate costs:
>     - The trace network is scalar-valued and demonstrates a much faster convergence compared to the main network, as shown in Fig. 1. In Laion-2B dataset, the main network of SD-1.5 needs 100,000 GPU hours, while the trace network only 3000, just **3%** of the main network's training cost.
>     - The trace network shares the same input as the diffusion network, allowing it to use a shared backbone of $\epsilon_\theta$ for feature extraction, and only necessitates the training of an output head. We plan to explore and implement this promising approach in our future research.
>
>     Thus, DF-TM can leverage off-the-shelf pretrained diffusion models and only needs to train a small trace network as a "plug".
>
> > **Question 1:**  Is it true that the prior methods all use the highly inefficient way of calculating the JVP...
>
> In prior adjoint methods [1][2], a JVP operation is required at each step. Usually, these methods use the official PyTorch auto-differentiation tools to compute the JVP.
> Our DF-EA approximation method circumvents the need for this JVP calculation, though the efficiency gain is limited.
>
> The main advantage of DF-EA in adjoint optimization lies in its accuracy, leading to better scores and visual effects, as demonstrated by the qualitative comparison in Fig. 4 and the quantitative comparison in Fig. 3.
>
> In the revision, we'll add more details on prior adjoint methods and their applications.
>
> [1] Pan, Jiachun, et al. "Towards accurate guided diffusion sampling through symplectic adjoint method." arXiv preprint arXiv:2312.12030 (2023).
>
> [2] Blasingame, Zander W., and Chen Liu. "AdjointDEIS: Efficient gradients for diffusion models." NeurIPS 2024.

---

> > ### Comment · Reviewer_mnN8 · 2025-04-04
> >
> > Thanks for the clarification. My concerns have been resolved, and I have raised my score accordingly.

---

> > > ### Author Response · Authors · 2025-04-05
> > >
> > > Dear Reviewer mnN8,
> > >
> > > Thank you very much for your thoughtful evaluation.  We are incredibly grateful for your discerning insight, reflecting your deep understanding and high standards.  Your positive assessment means a great deal to us.
> > >
> > > If you have any further suggestions for how we can improve our work and potentially improve your evaluation, please do let us know!
> > >
> > >
> > >
> > >
> > > Best regards! Wishing you a good day.

---

### Official Review · Reviewer_V5LL · 2025-03-16

**Overall Recommendation:** 2

**Summary:**

This paper addresses the challenge of efficiently accessing the diffusion Fisher information (DF) in diffusion models (DMs). Based on the analytical formulation of the diffusion Fisher, the authors propose two novel algorithms: DF Trace Matching (DF-TM) for efficiently estimating the trace of the DF, and DF Endpoint Approximation (DF-EA) for efficient matrix-vector multiplication with the DF. DF-TM trains a neural network to estimate the diffusion Fisher while DF-EA approximates the drift term through the learned score function. The paper provides theoretical error bounds for these approximations and showcases their superior accuracy and reduced computational cost in experiments on likelihood evaluation and adjoint optimization. Furthermore, leveraging their outer-product formulation, the authors show that their approach can be used to numerically verify the OT property of the diffusion-ODE derived map under various conditions.

## Update after rebuttal
- Most concerns have been clarified during the rebuttal.
- As an empirical likelihood estimation method, the proposed method showed improved accuracy based on low-dimensional experimental results. However, the practical significance of the method, based on the presented empirical evidence (Fig 2-4), remains somewhat unclear to me.
- The error bounds appear vacuous due to singularity. However, I admit that this issue originates from a fundamental limitation inherent to diffusion models—specifically, the singularity associated with the denoising score matching objective—rather than a flaw unique to the proposed method.
- Overall, I view this paper as borderline. However, considering its potential to inspire future research in the community, I would be supportive of acceptance if the other reviewers believe it's worth publishing.

**Claims And Evidence:**

Yes.

**Essential References Not Discussed:**

N/A

**Experimental Designs Or Analyses:**

- In the toy problem reported in Table 2, it is unclear how the ground truth NLL is obtained.

**Methods And Evaluation Criteria:**

Yes, it makes sense.

**Other Comments Or Suggestions:**

- Line 145: “outer-product sums We first” needs a period → “outer-product sums. We first”.
- Line 153: “in (Lu et al., 2022a)”. Please use inline citation when it is part of the sentence.
- Eq. (7) uses the second derivative to represent Hessian which is confusing. Please check out the standard the notation for [Hessian matrix](https://en.wikipedia.org/wiki/Hessian_matrix).

**Other Strengths And Weaknesses:**

**Strength**

- The proposed DF-TM is efficient and effective in terms of estimating the likelihood of an observed sample.
- The proposed method enables OT verification of diffusion models, which is an interesting application.

**Weaknesses**

- The bound in Proposition 5 is vacuous especially when t is small. NLL evaluation requires integration over [0, T] which means the overall approximation error bound of NLL will blow up.
- In the Dirac setting that the authors focus on, the direct likelihood evaluation becomes almost trivial. The likelihood of an observed $x$ is zero if $x$ is not one of the $y_i$;  is 1 / N i f $x$ is one of the $y_i$. Therefore, there is a gap from the typical challenges in generative modeling where the data distribution is general and we have only finite data samples. The paper should focus on the actual setting in the generative modeling.
- In the adjoint improvement task, the visual quality gain from proposed DF-EA adjoint is negligible.

**Questions For Authors:**

- The valuable setting in generative modeling is where we have finite samples from the general data distribution. Can the authors extend Proposition 4,5,6 for this setting?
- In the toy problem, how do you compute the ground truth NLL from the dataset?

**Relation To Broader Scientific Literature:**

This paper proposes a novel approach to estimate the diffusion Fisher.

**Theoretical Claims:**

Yes, I checked the proofs for Proposition 1 & 2.

---

> ### Author Rebuttal · Authors · 2025-03-27
>
> Thank you for your valuable feedback! We will answer your weaknesses/questions one by one.
>
> > **Weaknesses 1:** The bound in Proposition 5 is vacuous when t is small...
>
> It is true that as $t$ approaches $0$, the bound in Proposition 5 will be ill-defined and blow up due to division-by-zero. This concern is also raised by Reviewer a1EQ. We argue that this problem, which has been referred to as the **singularity issue** in previous literature [1][2], stems from the inherent characteristics of diffusion models rather than flaws in our derivations. A deep exploration of the singularity problem remains an open question, which we will investigate in our future works.
>
> Practically, we adopt the approach in [3], where the diffusion sampling path ended at $t_{\min} = 0.005$ instead of $t_{\min} = 0$. This circumvents potential blow-up effects of singularity issues and results in a well-defined bounded NLL.
>
> We briefly discussed this singularity issue on line 1242 of the original paper. In our revision, we will provide a more detailed discussion and clarification of our handling.
>
> [1] Zhang, P, et al. Tackling the singularities at the endpoints of diffusion models. CVPR 2024.
>
> [2] Yang, Z, et al. "Lipschitz singularities in diffusion models." ICLR 2024.
>
> [3] Song, Y, et al. Consistency Models ICML 2023.
>
>
> > **Weaknesses 2:** In the Dirac setting, the likelihood evaluation becomes trivial...
>
> We derive the theory within the Dirac setting to simplify the formulations and make them easier to follow. We also present the derivations under the general setting in the appendix. Remarkably, both the Dirac and general settings yield the same forms for the DF-TM and DF-EA methods.
>
> All derivations within the Dirac setting can be straightforwardly adapted to the general setting by replacing discrete summations with general integrals. The underlying ideas and mechanisms are identical.
> The Dirac case can be regarded as a collection of finite Monte-Carlo samples drawn from a general distribution. In the mean-field limit, the Dirac case converges to the general case.
>
> To prevent misunderstandings, we will revise all propositions in the main body of the paper to the general setting.
>
> > **Weaknesses 3:** In adjoint task, the visual quality...
>
> In Fig. 4, the DF-EA method contributes to (left) a higher Laion-Aes score, as evidenced by a smoother visual effect, and (right) a higher pick-score, reflected in enhanced colorfulness and more detailed features. Since visual effects can be subjective, we also present a quantitative demonstration of the score improvement brought about by DF-EA in Fig. 3.
>
>
> > **Suggestions:** Line 145...Line 153...Eq. (7)...
>
> We apologize for the oversights. We have now corrected the typographical errors and adopted standard Hessian notation for Eq. 7:
>  $\left(F_t(x_t,t)\right)_{i,j} := - \frac{\partial^2 \log q_t(x_t, t)}{\partial x_t^{(i)} \partial x_t^{(j)}}$, where $x^{(i)}$ denotes the i-th element of vector $x$.
>
> > **Question 1:** ...Can the authors extend Proposition 4,5,6 for this setting?
>
> Yes, our Proposition 3,4,5,6 can be extended to the general setting and result in **the same DF-TM and DF-EA method**. Suppose we have $m$ samples from a general initial distribution $q_0$.
>
> - **Proposition 3 (general version)**: *The trace of the diffusion Fisher for the diffused distribution $q_t$, where $t\in(0,1]$, is given by:
> $
> \mathrm{tr}\left(F_t(x_t,t)\right) = \frac{d}{\sigma_t^2} - \frac{\alpha_t^2}{\sigma_t^4}\left[ \int w(y) ||y||^2dq_0(y) - \left\|\int w(y) y dq_0(y)\right\|^2 \right]
> $*
>
> - **Proposition 4 (general version)**: *$\forall (x_t, t) \in \mathbb{R}^d \times \mathbb{R}^{+}$, the optimal $t_\theta(x_t, t)$ trained by Algo. 1 are equal to $\frac{1}{d}\int w(y) ||{y}||^2 dq_0(y)$.*
>
> - **Proposition 5 (general version)**: *The approximation error of DF-TM is at most $\frac{\alpha_t^2}{\sigma_t^4}\delta_1 +\frac{1}{\sigma_t^2}\delta_2^2$.*
>
> - **Proposition 6 (general version)**: *The approximation error of the DF-EA linear operator, is at most $\frac{\alpha_t^2}{\sigma_t^3}(2\mathcal{D}_y^2+ \sqrt{d}\delta_2)$*
>
> Note that, in Proposition 5 & 6, the approximation error induced from the finite number of $m$ samples is absorbed into $\delta_1$ and $\delta_2$.
> We'll include the general-version propositions and their proofs in the revision.
>
> > **Question 2:** In the toy problem, how do you compute the ground truth NLL from the dataset?
>
> In the toy experiment, we adopt the method in [1] and approximate the true NLL following Eq. 13. We use Euler discretization over 1000 timesteps for simulation and 5000 data samples per timestep to approximate the right-hand side term of Eq. 13. This setup aims to ensure NLL approximation accuracy.
>
> We apologize that the description in line ~1096 is rather simplistic. In the revision, we will provide a more comprehensive description.
>
> [1] Lu, Cheng, et al. Maximum likelihood training for score-based diffusion odes by high order denoising score matching. ICML 2022.

---

> > ### Comment · Reviewer_V5LL · 2025-04-07
> >
> > Thank you for the clarification. However, I believe further clarification on the singularity issue is still necessary.
> > - For the Dirac setting, the singularity at time 0 is inherent, which cannot be addressed as discussed in [1]. However, this is not the case for general data distributions with finite second moments.
> > - To illustrate concretely, consider a simple example where the data distribution is Gaussian $p_0(x) \sim \mathcal{N}(0, I)$ and linear noise schedule $\sigma(t)=t$. The score function at time t is $\nabla\log p_t(x_t)= - \frac{1}{1+t^2}x_t$, which is well-defined for all t. There is no singularity in this setting, even at 0. Therefore, it is unclear to me why the provided results of the general version (finite second moments) have singularity at time 0.
> > - Therefore, I would like to request the authors to write down the concrete derivations for the general setting and point out the source of singularity in the general setting. It would also be helpful if you could work through the Gaussian example above and indicate why (or why not) the singularity appears.
> > - Regarding the cited references,
> >     - The intrinsic singularity issue discussed in [1] appears only in the Dirac setting.
> >     - [2] only focuses on the singularity issue caused by specific noise schedule and network preconditioning.
> >     - Thus, neither reference fully accounts for the singularity behavior in the general finite-moment case discussed in this paper.
> > - BTW, Eq. (12) it appears to be a placeholder ("general setting") without an actual expression. Please correct this for completeness.
> >
> > [1] Zhang, P, et al. Tackling the singularities at the endpoints of diffusion models. CVPR 2024.
> >
> > [2] Yang, Z, et al. "Lipschitz singularities in diffusion models." ICLR 2024.

---

> > > ### Author Response · Authors · 2025-04-07
> > >
> > > Dear Reviewer V5LL
> > >
> > > We sincerely appreciate your profound reflection on the theory of diffusion models. Your concerns regarding the singularity issue of diffusion models demonstrate a deep understanding of this complex field, and we highly value your insights.​
> > >
> > > We will first conduct a comprehensive analysis of the two primary sources of the singularity problem in diffusion models when $t \to 0$. Subsequently, we will provide a detailed examination of your example.
> > >
> > > **Two sources of singularity**:
> > > - **Source 1: Discontinuity of the data distribution, leading to the ill-defined score $\nabla\log p_0(x_0)$.**
> > >     Due to the Radon–Nikodym theorem [1], $p_0$'s log-density function must be absolutely continuous (a.c.) with respect to the Lebesgue measure on $\mathbb{R}^d$ to ensure a well-defined score $\nabla\log p_0(x_0)$.
> > >
> > >     Notably, having finite second moments is not enough to guarantee that a distribution has a well-defined score. Distributions with finite second moments can still possess discontinuous components. Absolutely continuous property is needed to cut off this source of singularity.
> > >
> > >
> > > - **Source 2: The design of $\epsilon$-prediction of diffusion models parameterization, leading to divide-by-zero of $\sigma_0$.**
> > >     Current diffusion models generally do not use the network to directly match the score $\nabla\log p_t(x_t,t)$.
> > >     Instead, they use a network $\epsilon_\theta(x_t,t)$ to match a scaled version of the score, specifically $\epsilon^{\*}(x_t,t)=-\sigma_t\nabla\log p_t(x_t,t)$, which is known as $\epsilon$-prediction.
> > >
> > >     The score approximated by the learned network is then $\nabla\log p_t(x_t,t) =-\frac{\epsilon^{\*}(x_t,t)}{\sigma_t}\approx-\frac{\epsilon_\theta(x_t,t)}{\sigma_t}$.
> > >     The score $\nabla\log p_t(x_t,t)$ near $t=0$ can be theoretically obtained by calculating $\lim_{t\to 0}-\frac{\epsilon^{\*}(x_t,t)}{\sigma_t}$. However, both $\lim_{t\to 0}\epsilon^{\*}(x_t,t)=0$ and $\lim_{t\to 0}\sigma_t=0$, making this limitation hard to get.
> > >
> > >     Moreover, we also do not have access to $\epsilon^{\*}(x_t,t)$ in practice. We can only approximate it via $\epsilon_\theta(x_t,t)$.
> > >     Due to the network's matching error, $\epsilon_\theta(x_t,t)$ may not converge to $0$ as $t\to 0$.
> > >     This is precisely why calculating the score or other quantities using the learned $\epsilon_\theta(x_t,t)$ near $t=0$ leads to a blow-up divide-by-zero result.
> > >
> > >     Most existing off-the-shelf pretrained diffusion models, such as SD-15, SD-2base, SD-XL, and PixArt-$\alpha$ all adopt the $\epsilon$-prediction parameterization. Hence, our analysis on diffusion Fisher primarily focuses on this widely-adopted scenario and encounters the singularity issue.
> > >
> > > In practice, our DF-TM and DF-EA algorithms end the sampling path at $t_{\min}=0.005$, bypassing the singularity issue.
> > >
> > > [1] Folland G B. Real analysis: modern techniques and their applications[M]. John Wiley & Sons, 1999.
> > >
> > > > **Question 1:** consider a simple example where the data distribution is Gaussian $p_0(x) \sim \mathcal{N}(0, I)$ ...
> > >
> > > Your derivation is correct. When the data distribution is Gaussian, i.e., $p_0(x)\sim\mathcal{N}(0, I)$ and $\sigma_t = t$.
> > > The score is always well-defined as $\nabla\log p_t(x_t,t)=-\frac{1}{1 + t^2}x_t$. This analytical score has no singularity issue because $\lim_{t\to 0}\nabla\log p_t(x,t)=\lim_{t\to 0}-\frac{1}{1 + t^2}x=-x$ for any $x$.
> > >
> > > However, if we use $\epsilon$-prediction to learn this score, a troublesome singularity issue will occur. The $\epsilon$-prediction scaled score takes the form of $\epsilon^{\*}(x_t,t) = -\sigma_t\nabla\log p_t(x_t,t)=\frac{t}{1 + t^2}x_t$, which converge to $0$ as $t\to 0$.
> > > But the learned $\epsilon_{\theta}(x_t,t)$ will have matching errors compared to $\epsilon^{\*}(x_t,t)$ and will not be exactly $0$ as $t\to 0$.
> > > To estimate the score $\nabla\log p_t(x,t)$ as $t\to 0$ in practice, we can only calculate $\lim_{t\to 0} \frac{\epsilon_{\theta}(x_t,t)}{\sigma_t}$, which results in a divide-by-zero error.
> > >
> > > In summary, in this Gaussian case, the underlying analytical score has no singularity issue. However, attempting to obtain it through an $\epsilon$-prediction network gives rise to singularity issues. Unfortunately, in practice, we do not know the analytical score; we only know the learned $\epsilon$-prediction network.
> > >
> > >
> > > > **Question 2:** BTW, Eq. (12) it appears to be a placeholder ("general setting") without an actual expression...
> > >
> > > Thanks for your suggestion. We follow your kind advice to write out the detailed definition of the general finite second moments distributions as follows:
> > > $$
> > > q_0\in \mathcal{P}(\mathbb{R}^d),\quad \int_{\mathbb{R}^d} ||x||^{2} q_0(x)dx<\infty
> > > $$
> > > We will fix this in our revision.
> > >
> > >
> > >
> > > If Reviewer V5LL still has further concerns, we would appreciate it if you could **update them in the Rebuttal Comment**. We are more than willing to provide further detailed responses.
> > >
> > > **Wish you a happy day!**

---

### Decision · Program_Chairs · 2025-05-01

**Decision:**

Accept (poster)

**Comment:**

The paper addresses the challenge of efficiently accessing the diffusion Fisher information (DF) in diffusion models (DMs).
The authors derive an outer-product span formulation of the diffusion Fisher and introduce two approximation algorithms for different types of access to the diffusion Fisher: diffusion Fisher trace matching (DF-TM) and diffusion Fisher endpoint approximation (DF-EA).
Theoretical error bounds are provided, along with numerical experiments. Furthermore, using their outer-product formulation, the authors show that their approach can be used to numerically verify the optimal transport (OT) property of the diffusion-ODE map under various conditions.

Reviewers generally appreciate that the proposed approach is novel and interesting. Reviewers V5LL and a1EQ both share concerns regarding the singularity issue when t=0, in which case the theoretical bounds in Propositions 5 and 6 would be vacuous. In their rebuttal, the authors pointed out that this issue originates from a fundamental limitation inherent to diffusion models and not their proposed approach; Reviewer V5LL acknowledged this and Reviewer  a1EQ increased their score.

Overall, I believe that this would be an interesting contribution and recommend acceptance.